# FLAT-BENCH: A FEDERATED LEARNING BENCHMARK FOR ADAPTATION AND TRUST

## ABSTRACT

Federated Learning (FL) has emerged as a promising paradigm for collaborative model training while preserving data privacy across decentralized participants. As FL adoption grows, numerous techniques have been proposed to tackle its practical challenges. However, the lack of standardized evaluation across key dimensions hampers systematic progress and fair comparison of FL methods. In this work, we introduce **FLAT-Bench**, a unified framework for analyzing federated learning through two foundational dimensions: **A**daptation and **T**rust. We provide an in-depth examination of the conceptual foundations, task formulations, and open research challenges associated with each theme. We have extensively benchmarked representative methods and datasets for *adaptation to heterogeneous clients* and *trustworthiness in adversarial or unreliable environments*. **FLAT-Bench** lays the groundwork for systematic and holistic evaluation of federated learning with real-world relevance. We will make our complete codebase publicly accessible and a curated repository that continuously tracks new developments and research in the FL literature.

## 1 INTRODUCTION

Deep learning has revolutionized numerous fields, leading to groundbreaking advancements across various scientific domains, and has increasingly permeated industrial and societal applications. This transformation is especially evident in areas such as computer vision (Deng et al., 2009; Russakovsky et al., 2015a; Dosovitskiy et al., 2021; He et al., 2016; Xie et al., 2017; Huang et al., 2019; Yenduri et al., 2024), natural language processing (Vaswani et al., 2017; Devlin et al., 2019), multi-modal learning (Radford et al., 2021; Li et al., 2022; Zhang et al., 2023), and medical analysis (Chen et al., 2023c). With increasing concerns around data sensitivity and privacy, several regulatory frameworks have been introduced to regulate how data is collected and used (May & Sell, 2006; of Investigators for Fairness in Trial Data Sharing, 2016; Voigt & dem Bussche, 2017; Pardau, 2018). As a result, traditional centralized training approaches, which rely on aggregating raw data from multiple sources, face significant deployment challenges in real-world applications. To address these constraints, federated learning (FL) (Konečnỳ et al., 2016b;a; McMahan et al., 2017; Yang et al., 2019; Sun et al., 2020; Hong & Chae, 2021; Yang et al., 2021) has gained traction as an effective paradigm for privacy-aware collaborative learning. FL allows multiple participants to collaboratively train a shared model without sharing their data. Clients locally update the model using their data, and only the learned updates are sent to a central server, which aggregates them into a global model for further refinement. This decentralized approach allows FL to support secure and privacy-preserving learning across distributed data silos. Despite notable progress in FL research (Hard et al., 2018; Ju et al., 2020; Zhuang et al., 2020; Guo et al., 2020; Liu et al., 2020a; Wu & Gong, 2021; Pati et al., 2022; Chen et al., 2023b), the field still faces several open challenges. Two primary areas of concern are:

• **Adaptation**. In federated learning, data is often generated across diverse sources, naturally resulting in non-independent and non-identically distributed (Non-IID) characteristics (N.Shoham et al., 2019; N.Liu et al., 2019; K.Hsieh et al., 2020; T.Li et al., 2020a; X.Li et al., 2021; C.Wu et al., 2022; Y.Tan et al., 2023). These discrepancies introduce two primary types of distribution shifts: **i)** *Cross-Client Distribution Shift*: Each client typically operates on data with a distinct distribution, leading to significant heterogeneity between participants. As a result, clients tend to optimize their local models toward different empirical minima, which may conflict with one another (Q.Li et al., 2021a; M.Luo et al., 2021; L.Zhang et al., 2021b; Y.Dandi et al., 2022; Z.Qu et al., 2022). This misalignment

Table 1: **Summary of existing works**. Additional information can be found in Appendix C.

| Prior Works | Generalization | Adaptation | Trust | | |
|---|---|---|---|---|---|
| | | | Robustness | Fairness | Benchmark |
| [arXiv'18], Y.Zhao (2018), [TIST'19] Yang et al. (2019), [WS4'20] V.Kulkarni et al. (2020), [arXiv'21] L.Zhang et al. (2021a) [FGCS'22] X.Ma et al. (2022), [CSUR'23] M.Ye et al. (2023), [arXiv'23] Y.Li et al. (2023) [NC'21] H.Zhu et al. (2021), [CSUR'22] Nguyen et al. (2022a), [FGCS'22] X.Ma et al. (2022) | ✓ | | | | |
| [FGCS'21]V.Mothukuri et al. (2021), [SPM'20]T.Li et al. (2020c), [CSR'23]C.Xu et al. (2021) | ✓ | | ✓ | | |
| [FTML'21]P.Kairouz et al. (2021), [TKDE'21]Q.Li et al. (2021b) | ✓ | ✓ | ✓ | ✓ | |
| [arXiv'20]L.Lyu et al. (2020a), [TrustCom'22] J.Shi et al. (2022), [TNNLS'22] L.Lyu et al. (2022) | | | ✓ | | |
| [TKDE'21]Q.Li et al. (2021b), [arXiv'22] X.Liu et al. (2022), [arXiv'23] J.Shao et al. (2023) | ✓ | ✓ | | | ✓ |
| [TPAMI'24]Huang et al. (2024) [CVPR'24] Zhang et al. (2024) | ✓ | | ✓ | ✓ | ✓ |
| Ours | ✓ | ✓ | ✓ | ✓ | ✓ |

in optimization trajectories can hinder convergence and reduce the effectiveness of the aggregated global model. **ii)** *Out-of-Client Distribution Shift*: Federated models are trained solely on data from participating clients, and thus are biased toward the distributions present during training. When deployed in unseen environments or encountering new clients (*i.e.*, external domains), these models often underperform due to their inability to generalize beyond the observed training distributions (H.Yuan et al., 2022; X.Peng et al., 2020; Q.Liu et al., 2021; M.Jiang et al., 2023; L.Jiang & T.Lin, 2023). This issue limits the model's robustness in real-world scenarios.

• **Trust**. Although FL preserves privacy, its decentralized structure makes it vulnerable: a few compromised clients can poison local updates and skew global training. **i)** *Byzantine Attacks*: Clients may send malicious updates by poisoning local data (*data poisoning* (B.VanRooyen et al., 2015; B.Han et al., 2018)) or tampering with model weights (*model poisoning* (G.Baruch et al., 2019; C.Xie et al., 2020b; M.Fang et al., 2020)), degrading model accuracy. **ii)** *Backdoor Attacks*: Adversaries embed triggers in their updates so the global model misclassifies specific inputs while appearing normal otherwise (X.Chen et al., 2017; C.Liao et al., 2018; T.Gu et al., 2019). Distributed trigger schemes further evade detection by splitting patterns across clients (C.Xie et al., 2020a; X.Lyu et al., 2023). In high-stakes applications such as medical imaging (Nguyen et al., 2022a), autonomous driving (A.Nguyen et al., 2022), and fraud detection (W.Zheng et al., 2021), these threats demand robust defenses and fair reward mechanisms to ensure long-term collaboration. **iii)** *Privacy-Preserving Adaptation:* Adapting pretrained models to local tasks (*e.g.*, via federated fine-tuning methods such as LoRA (Hu et al., 2021)) must preserve data privacy while maintaining robustness under heterogeneous client objectives (Li et al., 2020).

Despite growing interest in adaptation and trust, the absence of a unified evaluation framework limits systematic progress. We address this by introducing a structured benchmark that consolidates these challenges for robust, comparative assessment. As shown in Table 1, prior works often focus on isolated FL challenges *e.g.*, generalization (Y.Zhao, 2018), robustness (L.Lyu et al., 2020a), or fairness (Y.Shi et al., 2023a) without offering unified perspectives. In contrast, our benchmark holistically evaluates adaptation and trust (robustness and fairness) making our contributions threefold:

• We introduce **FLAT-Bench**, a unified benchmark that not only categorizes key federated learning challenges across **A**daptation and **T**rustworthiness, but also formalizes task settings, evaluation criteria, and research gaps in current literature.
• We conduct extensive empirical evaluations covering adaptation and trust (robustness and fairness) across diverse FL settings.
• We highlight future research directions and consolidate key datasets, tasks, and method trends to guide actionable progress in federated learning deployments.

## 2 ADAPTIVE FEDERATED LEARNING

Adaptive Federated Learning tackles generalization and personalization across diverse clients. It balances global performance with client-specific adaptation using techniques like meta-learning and fine-tuning, enabling effective deployment in Non-IID settings such as healthcare and cross-device systems.

**Cross Calibration.** In the case of *Cross-Client Shift* challenge, client data is often distributed in a highly skewed manner, which results in inconsistencies between local training goals. Consequently, each client updates its model based on a distinct local optimum, resulting in divergence of optimization

Table 2: **Overview of Key Attributes in Reviewed Techniques for Cross Calibration** (See § 2).

| Method | Venue | Core Idea |
|---|---|---|
| *Global Neural Network* | | |
| **Drawback**: Linear growth in local computational load | | |
| FedProx T.Li et al. (2020a) | [MLSys'20] | $\ell_2$-based constraint on updates |
| SCAFFOLD Karimireddy et al. (2020) | [ICML'20] | Gradient correction via control variates |
| MOON Q.Li et al. (2021a) | [CVPR'21] | Contrastive learning in feature space |
| FedNTD G.Lee et al. (2022) | [NeurIPS'22] | Decoupled approach to knowledge transfer |
| FedSeg Miao et al. (2023) | [CVPR'23] | Contrastive strategy at pixel-level granularity |
| GeFL Kang et al. (2024) | [arXiv'24] | Aggregate global knowledge across users |
| *Global Statistical Cues* | | |
| **Drawback**: Heavily dependent on comprehensive data diversity | | |
| FedProc X.Mu et al. (2021) | [arXiv'21] | Use of prototype similarity for contrast |
| HarmoFL M.Jiang et al. (2022) | [AAAI'22] | Employs signal amplitude normalization |
| FedFA T.Zhou & E.Konukoglu (2023) | [ICLR'23] | Data augmentation via Gaussian modeling |
| FPL Huang et al. (2023) | [CVPR'23] | Prototype refinement using clustering |
| FedSB Soltany et al. (2025) | [ICASSP'25] | Utilizes label smoothing to prevent overfitting |
| *Augmented Architectures* | | |
| **Drawback**: Introduces integration issues and added overhead | | |
| FedML B.J.Kim et al. (2022) | [ICML'22] | Multi-branch architecture for flexibility |
| FedCGAN Y.Wu et al. (2022) | [IJCAI'22] | GAN-based synthetic data generation |
| ADCOL Q.Li et al. (2023) | [ICML'23] | Generator that learns client representations |
| DaFKD H.Wang et al. (2023) | [CVPR'23] | Introduces a discriminator for distillation |
| CAFA Kouda et al. (2025) | [FGCS'25] | Leverages computational capacities for local training |
| *Self-Regulated Learning* | | |
| **Drawback**: Hyperparameter tuning instability, risk of forgetting | | |
| FedRS Li & Zhan (2021) | [KDD'21] | Limits softmax confidence levels |
| FedAlign M.Mendieta et al. (2022) | [CVPR'22] | Ensures final layer stability via Lipschitz constraints |
| FedSAM Z.Qu et al. (2022) | [ICML'22] | Applies sharpness-aware optimization |
| FedLC Zhang et al. (2022) | [ICML'22] | Adjusts logits using class-wise probability |
| FedDecorr Y.Shi et al. (2023b) | [ICLR'23] | Reduces inter-feature redundancy |
| FedVR-AL Thakur et al. (2024) | [arXiv'24] | Variance reduction and adaptation for non-convex optimization |

| Method | Venue | Core Idea |
|---|---|---|
| *Collaborative Data Sharing* | | |
| **Drawback**: Assumes prior availability of suitable external data | | |
| DC-Adam P.Tian et al. (2021) | [CS'21] | Initial warm-up using pre-distributed data |
| FEDAUX F.Sattler et al. (2021) | [TNNLS'21] | Auxiliary data for pretraining and distillation |
| ProxyFL Kalra et al. (2023) | [NatureComms'23] | Shares proxy models across clients |
| ShareFL Shao et al. (2024) | [arXiv'23] | Review on collaborative data sharing in FL |
| FedSPD Lin et al. (2024) | [arXiv'24] | Clustering-based framework enabling consensus for distinct data clusters |
| *Data Augmentation for FL* | | |
| **Drawback**: May reduce data variety, can cause privacy issues | | |
| FedMix T.Yoon et al. (2021) | [ICLR'21] | Mixup of averaged samples across clients |
| FEDGENZ.Zhu et al. (2021) | [ICML'21] | Uses ensemble generators for diversity |
| FedInverse Wu et al. (2024) | [ICLR'24] | Investigates inversion attacks and defenses |
| FLea Xia et al. (2024) | [KDD'24] | Privacy-preserving feature augmentation techniques |
| *Sample Filtering in FL* | | |
| **Drawback**: Risk of unfair exclusion at client/data level | | |
| FedACS Wang et al. (2021) | [IWQOS'21] | Detects and excludes poisoned data via clustering |
| SafeX X.Xu et al. (2022) | [TII'22] | Prefers clients with lower distributional skew |
| FedBalancer Shin et al. (2022) | [MobiSys'22] | Prioritizes fair data sampling across devices |
| Fedrtid Yang et al. (2024) | [Cybersecurity'24] | Introducing random client participation and adaptive time constraints |
| *Aggregation Reweighting at Server* | | |
| **Drawback**: Requires thorough dataset quality evaluation | | |
| FEDBE Chen & Chao (2021) | [ICLR'21] | Uses Bayesian ensembles for aggregation |
| Elastic Dengsheng et al. (2023) | [CVPR'23] | Aggregates via parameter sensitivity interpolation |
| FFA Dilley et al. (2024) | [arXiv'24] | Novel metrics that consider client participation and aggregation methods |
| *Server-Side Adaptive Methods* | | |
| **Drawback**: Needs auxiliary data and aligned training objectives | | |
| FedMD Li & Wang (2019) | [NeurIPS'19] | Distills from local classifiers on proxy data |
| FedDF Lin et al. (2020) | [NeurIPS'20] | Combines knowledge from diverse client models |
| FedGKT He et al. (2020) | [NeurIPS'20] | Shares group knowledge across clients |
| FedOPT Reddi et al. (2021) | [ICLR'21] | Adaptive optimization on central server |
| FCCL Huang et al. (2022) | [CVPR'22] | Cross-correlation for representation alignment |

directions. Existing approaches primarily aim to mitigate this divergence by adjusting client updates from three key perspectives, as shown in Table 2.

**Client Regularization.** Federated methods that seek to align client updates with a shared global objective can be broadly classified into four categories. First, global neural network guidance directly incorporates the aggregated model into each client's local update either via parameter-sensitivity constraints (*e.g.*, FedProx (T.Li et al., 2020a), FedCurv (N.Shoham et al., 2019), FedDyn (Acar et al., 2021)) or by penalizing divergence from global predictions (*e.g.*, MOON (Q.Li et al., 2021a), FedUFO (L.Zhang et al., 2021b)) at the cost of increased computation that scales with model size. Second, global statistical cues approaches construct class-wise summaries (*e.g.*, prototypes (X.Mu et al., 2021), Gaussian descriptors (M.Luo et al., 2021), spectral signatures (M.Jiang et al., 2022)) or aggregate feature representations (Peng et al., 2022) to provide finer-grained guidance, though their reliability depends on the diversity and richness of client data. Third, augmented architectures introduce supplementary modules such as GAN-based generators (Z.Zhu et al., 2021; H.Wang et al., 2023) or parallel "global" branches (He et al., 2020; J.Kim et al., 2022) to counter client drift, but these often require architectural compatibility and increase communication overhead. Finally, self-regulated learning leverage self-distillation (Yu et al., 2021) or reweighted loss functions (Li & Zhan, 2021; Y.Shi et al., 2023b) to stabilize local training without extra communication, though their effectiveness can be highly sensitive to hyperparameters, especially under extreme data heterogeneity.

**Client Augmentation.** To mitigate client data heterogeneity, FL methods can be broadly grouped into three strategies. First, collaborative data sharing exchanges labeled or unlabeled examples or models among clients to promote knowledge transfer. Approaches like DC-Adam (P.Tian et al., 2021) and FEDAUX (F.Sattler et al., 2021) use warm-up phases or auxiliary pretraining, while others like ProxyFL (Kalra et al., 2023) share proxy models to enable indirect data knowledge exchange. ShareFL (Shao et al., 2024) provides a comprehensive review, and FedSPD (Lin et al., 2024) enables inter-client clustering to reach consensus among data-similar clients. However, these strategies assume the availability of meaningful and appropriately matched auxiliary data, which may not always be feasible. Second, data augmentation enhances local datasets to simulate more diverse conditions. Methods like FedMix (T.Yoon et al., 2021) mix local data representations across clients, FEDGEN (Z.Zhu et al., 2021) employs ensemble generators to synthesize informative samples, and FedInverse (Wu et al., 2024) explores the privacy implications of such augmentations. FLea (Xia et al., 2024) applies privacy-preserving feature augmentation techniques. While useful, these methods can reduce diversity or inadvertently leak private data through reconstruction or overfitting. Third, sample filtering avoids direct data sharing or augmentation by selecting clients or samples deemed more trustworthy. For example, FedACS (Wang et al., 2021) and Safe (X.Xu et al., 2022) cluster data or prioritize lower-skew clients, respectively. FedBalancer (Shin et al., 2022) balances fairness by allocating sampling quotas, and Fedrtid (Yang et al., 2024) introduces random client participation with adaptive timing to reduce resource burden and enhance robustness. However, these methods risk marginalizing clients with less "mainstream" data, undermining fairness.

Table 3: **Overview of key properties of the evaluated methods for Unknown Generalization (see § 2)**. The symbols $\star$ and $\circ$ indicate **possible privacy exposure** and **modifications to the model architecture**, respectively.

| Federated Domain Adaptation | | | |
|---|---|---|---|
| Methods | Venue | Highlight | Limitation |
| FADA X.Peng et al. (2020) | [ICLR'20] | Adversarial alignment | ∘: Uses GAN Goodfellow et al. (2014) |
| COPA G.Wu & S.Gong (2021) | [ICCV'21] | Shared encoder, task heads | ∘: Needs IBN X.Pan et al. (2018) |
| AEGR G.Li et al. (2023) | [ICME'23] | Pseudo-label tuning | ★: Exposed to PGD A.Madry et al. (2017) |
| FedGP Dai et al. (2024) | [ICLR'24] | Gradient projection aggregation | Requires projection tuning |
| FedRF-TCA Feng et al. (2025) | [TKDE'25] | Random features for efficiency | May underperform on complex domains |

| Federated Domain Generalization | | | |
|---|---|---|---|
| Methods | Venue | Highlight | Limitation |
| FedDG Q.Liu et al. (2021) | [CVPR'21] | Frequency-based sharing | ★: Reveals amplitude |
| CCST Chen et al. (2023a) | [WACV'23] | Client-wise style mixing | ★: Leaks style cues |
| CSAC J.Yuan et al. (2023) | [TKDE'23] | Semantic layer fusion | ∘: Adds attention |
| FedSB Soltany et al. (2025) | [ICASSP'25] | Label smoothing and balanced training | Careful tuning of smoothing parameters |
| FedCGA Liu et al. (2024b) | [ICME'24] | Global consistent augmentation | Assumes availability of diverse styles |

**Server Operation.** To better handle heterogeneous client updates, federated learning can adapt aggregation dynamics at the server. One direction is aggregation reweighting, where clients are weighted based on factors beyond static proportions. For instance, FEDBE (Chen & Chao, 2021) uses Bayesian ensembling, Elastic (Dengsheng et al., 2023) reweights updates using gradient sensitivity, and FFA (Dilley et al., 2024) introduces fairness-aware metrics to evaluate participation and aggregation impacts. While these improve personalization and convergence, they rely on costly evaluations of data quality or model variance. A complementary direction is server-side adaptive optimization, where the central model is refined using external data or tailored learning rules. Methods like FedMD (Li & Wang, 2019), FedDF (Lin et al., 2020), and FedGKT (He et al., 2020) distill knowledge across clients using proxy data. FedOPT (Reddi et al., 2021) adapts server-side optimization rules, while FCCL (Huang et al., 2022) aligns representations using cross-correlation signals. Though effective, such approaches often require additional datasets and tuned objectives, which may complicate real-world deployment.

**Unknown Generalization.** Prior studies have shown that deep neural networks often overfit their training data and produce overly confident outputs (C.Guo et al., 2017; B.Lakshminarayanan et al., 2017b). We summarize the essential characteristics of various solutions addressing Unknown Generalization in Table 3. Such overconfidence can prove detrimental in practice (D.Amodei et al., 2016), as even slight distributional shifts between training and deployment data may lead to substantial performance degradation (B.Lakshminarayanan et al., 2017a; Y.Ovadia et al., 2019). In federated learning, the majority of the work concentrates on boosting in-distribution accuracy across clients, with limited attention paid to how models generalize to novel, out-of-federation domains (D.Peterson et al., 2019; X.Peng et al., 2020; Q.Liu et al., 2021; H.Yuan et al., 2022). Approaches addressing this gap can be categorized according to when they gain access to out-of-distribution data: Federated Domain Adaptation (FDA) and Federated Domain Generalization (FDG). FDA methods incorporate unlabeled target-domain samples during training to reduce distribution shift, and can be broadly categorized into alignment-based approaches which enforce feature consistency through contrastive losses (Y.Wei et al., 2022; Y.Wei & Y.Han, 2023), knowledge-distillation alignment (H.Feng et al., 2021; Z.Niu et al., 2023; X.Liu et al., 2023), adversarial adaptation (G.Li et al., 2023), or gradient matching (Zhu et al., 2022; Zeng et al., 2022) and disentanglement-based methods, which split the model into shared and domain-specific components via adversarial losses (X.Peng et al., 2020; L.Huang et al., 2011), multi-expert gating (Zec et al., 2020), or separate classifiers (G.Wu & S.Gong, 2021). In contrast, FDG seeks to train on heterogeneous client data and generalize directly to unseen domains, using either invariant optimization techniques, such as spectrum alignment (Q.Liu et al., 2021), style normalization (Chen et al., 2023a), barycenter-based feature fusion (Zhou et al., 2023), or specialized architectural blocks (GANs (L.Zhang et al., 2021a), AdaIN (Chen et al., 2023a), IBN (G.Wu & S.Gong, 2021)) or invariant aggregation schemes that reweight or calibrate server-side model fusion to balance domain performance (R.Zhang et al., 2023; Duan et al., 2023; J.Yuan et al., 2023).

## 3 TRUSTWORTHY FEDERATED LEARNING

Trustworthy Federated Learning centers on **robustness** and **fairness**. Robustness addresses threats from adversarial clients or corrupted updates, while fairness ensures equitable performance across heterogeneous users. Together, they define the trust boundary essential for FL deployment in sensitive domains like healthcare and finance.

**Byzantine Tolerance.** To guard against *Byzantine* clients, robust aggregation methods can be grouped into three families: distance-based tolerance, which detects and discards updates that deviate strongly from the group consensus (*e.g.*, Krum (Blanchard et al., 2017), FoolsGold (Fung et al., 2018), FABA (Q.Xia et al., 2019)); statistical-based tolerance, which applies robust estimators such as the

geometric median or trimmed means to filter outliers without tracking individual contributions (*e.g.*, RFA (K.Pillutla et al., 2022), Bulyan (R.Guerraoui et al., 2018)); and proxy-based tolerance, which uses a small, clean auxiliary dataset to score and weight client updates by their performance on trusted samples (*e.g.*, Sageflow (J.Park et al., 2021), FLTrust (X.Cao et al., 2021b)). Similarly, mitigating backdoor attacks has led to three main defense paradigms: post-hoc model sanitization, where the aggregated model is fine-tuned or distilled on clean data to erase backdoors (*e.g.*, FedPurning (C.Wu et al., 2020), FedDF (Lin et al., 2020)); aggregation-time filtering, which extends Byzantine defenses to remove poisoned updates during server aggregation (*e.g.*, DimKrum (Z.Zhang et al., 2022), RLR (Ozdayi et al., 2021)); and certified defenses, which construct provable guarantees by maintaining multiple model variants or applying randomized smoothing so that small client perturbations cannot alter predictions (*e.g.*, ProvableFL (X.Cao et al., 2021a), CRFL (C.Xie et al., 2021)). Each category trades off different assumptions, computational costs, and requirements for auxiliary data or statistical priors, and their effectiveness can degrade significantly under real-world heterogeneity. Table 4 summarizes the essential characteristics of Byzantine Tolerance solutions discussed above.

**Collaboration Fairness.** In federated learning, fair contribution evaluation is critical to reward clients in proportion to their inputs while respecting data privacy (L.Lyu et al., 2020d;b). A common strategy is individualized evaluation, where each client's score is derived from locally available signals such as data acquisition cost (J.Zhang et al., 2020), economic incentives (*e.g.*, contract theory (J.Kang et al., 2019), Stackelberg models (M.Simaan & Cruz, 1973)), compute bids (Thi Le et al., 2021), or performance-based reputations computed via local validation (L.Lyu et al., 2020c) or update divergence from the global model (Li et al., 2021). However, this approach assumes honest reporting and can penalize clients with non-IID or smaller datasets. An alternative is marginal contribution estima-

Table 4: **Key characteristics of the reviewed Byzantine Tolerance solutions** as discussed in (§ 3).

| Methods | Venue | Highlight |
|---|---|---|
| *Distance Base Tolerance* | | |
| **Limitation**: Poor handling of data heterogeneity | | |
| Multi Krum Blanchard et al. (2017) | [NeurIPS'17] | Selects gradients using Krum rule |
| FoolsGold Fung et al. (2018) | [arXiv'18] | Detects sybils via similarity scores |
| DnC Shejwalkar & Houmansadr (2021) | [NDSS'21] | Uses SVD to isolate abnormal updates |
| RED-FL Herath et al. (2023) | [GlobConET'23] | Distance-based method to assign weights to client updates |
| FedWad Rakotomamonjy et al. (2024) | [ICLR'24] | Compute Wasserstein distances |
| *Statistics Distribution Tolerance* | | |
| **Limitation**: Depends on strong mathematical assumptions | | |
| Trim Median D.Yin et al. (2018) | [ICML'18] | Applies trimmed mean per dimension |
| Bulyan R.Guerraoui et al. (2018) | [ICML'18] | Selects top vectors, aggregates per axis |
| RFA K.Pillutla et al. (2022) | [TSP'22] | Iterative median via Weiszfeld approach |
| OPDS-FL Liu et al. (2023b) | [NeurIPS'23] | Measure data heterogeneity across clients |
| DFL-FS Chen et al. (2024) | [ICME'24] | Address long-tailed and non-IID data distributions |
| FD-PerFL Mclaughlin & Su (2024) | [NeurIPS'24] | Feature distributions for personalized federated learning |
| *Proxy Dataset Tolerance* | | |
| **Limitation**: Needs trusted data and client similarity | | |
| FLTrust X.Cao et al. (2021b) | [NDSS'21] | Uses trusted seed and ReLU score |
| Sageflow J.Park et al. (2021) | [NeurIPS'21] | Adjusts weights via entropy and loss |
| ProxyZKP Li et al. (2024) | [ScientificReports'24] | Zero-knowledge proofs with polynomial proxy models |

tion via cooperative game theory, notably Shapley value approximations (Shapley, 1997; Garrido-Lucero et al., 2024; X.Xu et al., 2021). Methods like Cosine-Gradient Shapley (CGSV) (X.Xu et al., 2021) and FEDCE (Jiang et al., 2023) evaluate each client's impact on model performance, but suffer from exponential complexity and often require auxiliary validation data, limiting their scalability in large-scale federations.

**Performance Fairness.** Performance imbalance in federated learning arises when the global model disproportionately favors clients with abundant or homogeneous data, leaving underrepresented participants with subpar accuracy. To mitigate this, two main classes of methods have emerged: (i) fairness-aware optimization, which embeds fairness constraints directly into each client's local loss—for example, min–max formulations such as AFL (M.Mohri et al., 2019) and loss-penalizing schemes like qFFL (T.Li et al., 2020b), or multi-objective descent approaches such as FedMGDA (Z.Hu et al., 2020) and FCFL (Cui et al., 2021) to uplift the worst-performing clients; and (ii) fair aggregation reweighting, which dynamically adjusts server-side combination weights based on client-level signals (*e.g.*, gradient conflict in FedFV (Z.Wang et al., 2021) or variance of generalization gaps in FedCE (Jiang et al., 2023; Ezzeldin et al., 2023)). While optimization-based strategies can improve the tail accuracy, they often assume honest reporting and can degrade overall utility; reweighting methods reduce skew via stale or auxiliary risk estimates, but incur extra synchronization overhead and may require validation data.

# 4 BENCHMARK SETUP

**Label Skew Datasets.** A common approach in current studies to emulate Label Skew scenarios involves using the Dirichlet distribution, denoted as $Dir(\beta)$ (Appendix A.2.1), for experimental purposes (Li et al., 2018; 2021). In this context, $\beta > 0$ acts as a concentration parameter that dictates the extent of class imbalance. Smaller values of $\beta$ cause a sharper disparity between local and global class distributions, intensifying data heterogeneity among clients. • **Cifar-10** (Krizhevsky et al.,

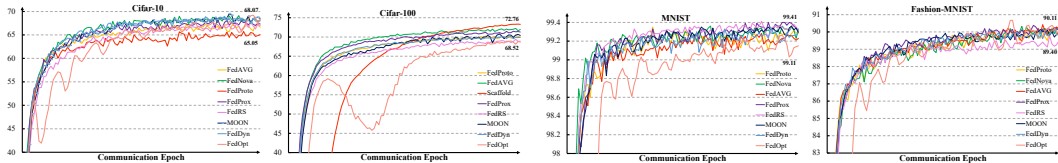

Figure 1: Test accuracy over 100 communication rounds on Cifar-10, Cifar-100, MNIST, and Fashion-MNIST datasets under Dirichlet distribution with $\beta = 0.5$.

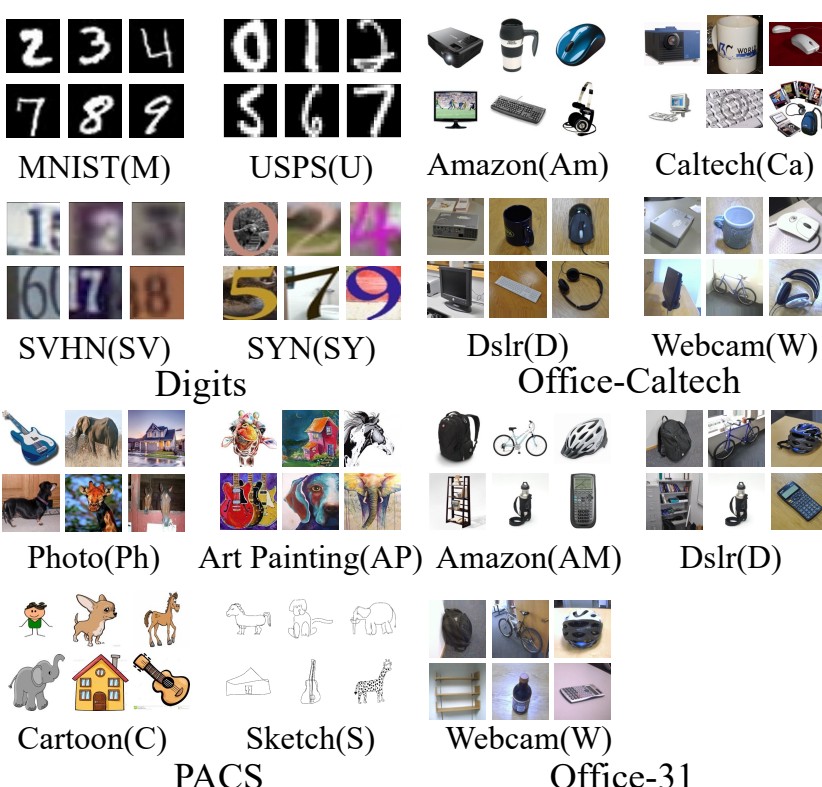

Figure 2: Visualization for Digits (Y.LeCun et al., 1998; Hull, 1994; Y.Netzer et al., 2011; Kingma & Welling, 2013), Office Caltech (Fei-Fei et al., 2007), PACS , and Office31 (Saenko et al., 2010). Refer to § 4.

2009) contains 50,000 images for training and 10,000 images for the validation. Its image size is 32 × 32 within 10 categories. ● **Cifar-100** (Krizhevsky et al., 2009) is a famous image classification dataset, containing 32 × 32 images of 100 categories. Training and validating sets are composed of 50,000 and 10,000 images. ● **Tiny-ImageNet** (Russakovsky et al., 2015b) is the subset of ImageNet with 100K images of size 64 × 64 with 200 classes scale. ● **Fashion-MNIST** (Xiao et al., 2017) includes 70,000 28 × 28 grayscale fashion product images with ten categories. Figure 1 illustrates test accuracy over 100 communication rounds for various federated learning methods on Cifar-10, Cifar-100, MNIST, and Fashion-MNIST under a Dirichlet distribution with $\beta = 0.5$. Figure 2 provides an overview of the datasets involved.

**Domain Skew & Out-Client Shift Datasets.** Both Domain Skew and Out-Client Shift scenarios involve datasets originating from different domains, where the main distinction lies in how evaluation is conducted. In Domain Skew, each client has domain-specific feature variations, as described in Appendix A.2.1. In contrast, Out-Client Shift adopts a leave-one-domain-out evaluation strategy, where one domain is treated as the unseen target client and the remaining domains are used collectively as sources for training. Examples from federated domain datasets are illustrated in Figure 2. ● **Office Caltech** combines samples from the Office dataset and Caltech256 (Fei-Fei et al., 2007), focusing on

Table 5: **Performance under Label Skew conditions** on Cifar-10, Cifar-100, MNIST, and Fashion-MNIST datasets, measured using $\mathcal{A}^{\mathcal{U}}$, and $\mathcal{E}$ (with $\beta = 0.5$) as defined in Appendix A.2.1. Bold indicates the highest value, underline marks the second-best, and "/" denotes zero or NaN. See Appendix E.1 for metric details and Appendix E.3 for further insights.

| Methods | Cifar-10 | | | | | Cifar-100 | | | | | MNIST | | | | | Fashion-MNIST | | | | |
|---|---|---|---|---|---|---|---|---|---|---|---|---|---|---|---|---|---|---|---|---|
| | 1.0 | 0.5 | 0.3 | 0.1 | $\mathcal{E}$ | 1.0 | 0.5 | 0.3 | 0.1 | $\mathcal{E}$ | 1.0 | 0.5 | 0.3 | 0.1 | $\mathcal{E}$ | 1.0 | 0.5 | 0.3 | 0.1 | $\mathcal{E}$ |
| FedAvg McMahan et al. (2017) | 70.64 | 66.96 | 63.92 | 60.43 | 0.354 | 68.47 | 69.72 | 69.21 | 68.92 | 0.213 | 99.44 | 99.37 | 99.13 | 98.76 | 0.602 | 89.94 | 89.87 | 83.82 | **90.15** | 0.462 |
| FedProx T.Li et al. (2020a) | **71.22** | 67.16 | 64.88 | 61.03 | 0.423 | **72.37** | 70.19 | 63.48 | 67.4 | 0.773 | 99.15 | **99.41** | **99.32** | 98.73 | 0.114 | 89.87 | 89.97 | 88.69 | 83.57 | 0.524 |
| SCAFFOLD Karimireddy et al. (2020) | 70.77 | **68.33** | **68.34** | 60.83 | / | 71.91 | **72.76** | 69.82 | **68.24** | / | 99.41 | 99.12 | 98.95 | 96.95 | / | 89.83 | 89.73 | 88.32 | 81.27 | / |
| FedNova Wang et al. (2020) | 70.94 | 67.06 | 66.42 | **64.05** | / | 70.12 | 67.11 | 63.86 | 27.91 | / | 99.42 | 99.29 | 99.22 | **99.88** | / | 90.20 | 89.81 | **89.03** | 84.39 | / |
| MOON Q.Li et al. (2021a) | 69.73 | 68.07 | 66.48 | 61.71 | 0.063 | 71.47 | 69.51 | 69.09 | 65.53 | 0.412 | **99.51** | 99.36 | 99.17 | 98.02 | 0.324 | **90.52** | **90.11** | 88.95 | 82.92 | 0.614 |
| FedRS Li & Zhan (2021) | 70.14 | 66.036 | 63.89 | 59.47 | 0.184 | 69.81 | 68.53 | 67.32 | 67.16 | 0.637 | 99.34 | 99.33 | 99.23 | 98.93 | 0.333 | 90.01 | 89.40 | 88.47 | 77.54 | 0.579 |
| FedDyn Acar et al. (2021) | 70.59 | 67.80 | 64.39 | 60.52 | 0.488 | 71.48 | 71.25 | 70.28 | 66.81 | 0.583 | 99.48 | 99.31 | 99.10 | 98.71 | 0.059 | 90.24 | 89.97 | 88.59 | 82.92 | 0.533 |
| FedOPT Reddi et al. (2021) | 70.44 | 66.70 | 65.95 | 63.10 | / | 69.40 | 68.52 | 67.57 | 67.26 | / | 99.32 | **99.11** | 98.92 | 98.13 | / | 90.06 | 89.65 | 88.79 | 83.41 | / |
| FedProto Tan et al. (2022) | 69.75 | 65.05 | 56.45 | 48.74 | 0.319 | 70.07 | 70.83 | 68.32 | 67.36 | 0.759 | 99.44 | 99.26 | 99.12 | 98.69 | 0.323 | 90.17 | 90.07 | 88.73 | 83.26 | 0.444 |
| FedNTD G.Lee et al. (2022) | 51.43 | 35.06 | 37.37 | 22.18 | 0.647 | 32.48 | 28.92 | 24.36 | 21.21 | 0.492 | 85.47 | 31.41 | 78.87 | 30.18 | 0.930 | 83.67 | 79.23 | 70.12 | 52.04 | 0.782 |

Table 6: **Quantitative Domain Skew results** in term of $\mathcal{A}^{\mathcal{U}}$, $\mathcal{A}^{u}$, $\mathcal{E}$, and $\mathcal{V}$ (Defined in E.3) on Digits, Office Caltech, and PACS. Refer to § 4.1.

| Methods | Digits | | | | | | | Office Caltech | | | | | | | PACS | | | | | | |
|---|---|---|---|---|---|---|---|---|---|---|---|---|---|---|---|---|---|---|---|---|---|
| | M | U | Svz | Sy | $\mathcal{A}^{\mathcal{U}}$ | $\mathcal{E}$ | $\mathcal{V}$ | Am | Ca | D | W | $\mathcal{A}^{\mathcal{U}}$ | $\mathcal{E}$ | $\mathcal{V}$ | P | AP | Ct | Sk | $\mathcal{A}^{\mathcal{U}}$ | $\mathcal{E}$ | $\mathcal{V}$ |
| FedAvg McMahan et al. (2017) | 90.40 | 60.30 | 34.68 | 46.99 | 58.09 | 0.024 | 4.35 | 81.99 | 73.21 | 79.37 | 67.93 | 75.62 | 0.653 | 0.379 | 76.09 | 64.19 | 83.50 | 89.40 | 78.30 | 0.279 | 0.911 |
| FedProx T.Li et al. (2020a) | 95.03 | 63.25 | 34.50 | 44.60 | 59.34 | 0.059 | 5.44 | 85.26 | 75.08 | 84.67 | 75.17 | 80.23 | 0.717 | 0.273 | 79.26 | 69.86 | 80.51 | 90.82 | 80.19 | 0.170 | 0.612 |
| SCAFFOLD Karimireddy et al. (2020) | 97.79 | 94.45 | 26.64 | 90.69 | 77.39 | / | 8.93 | 39.79 | 42.50 | 78.02 | 70.69 | 57.75 | / | 0.281 | 61.95 | 45.44 | 58.87 | 54.64 | 55.25 | / | 0.383 |
| MOON Q.Li et al. (2021a) | 92.78 | 68.11 | 33.36 | 39.28 | 58.36 | 0.287 | 5.72 | 84.42 | 75.98 | 84.67 | 68.97 | 78.51 | 0.678 | 0.539 | 74.44 | 64.19 | 83.92 | 89.17 | 77.93 | 0.321 | 0.924 |
| FedDyn Acar et al. (2021) | 88.91 | 60.34 | 34.57 | 50.72 | 58.65 | 0.161 | 4.06 | 84.02 | 72.59 | 77.34 | 68.97 | 75.72 | 0.824 | 0.430 | 78.17 | 64.29 | 82.27 | 89.93 | 78.66 | 0.129 | 0.881 |
| FedOPT Reddi et al. (2021) | 92.71 | 87.62 | 31.32 | 87.92 | 74.89 | / | 6.37 | 79.05 | 71.96 | 89.34 | 74.48 | 78.71 | / | 0.480 | 78.66 | 67.66 | 82.41 | 83.68 | 78.12 | / | 0.410 |
| FedProto Tan et al. (2022) | 90.54 | 89.54 | 34.61 | 58.00 | 68.18 | 0.558 | 5.47 | 87.79 | 75.98 | 90.0 | 79.31 | **83.27** | 0.556 | 0.410 | 85.63 | 73.69 | 83.57 | 91.14 | **83.51** | 0.540 | 0.411 |
| FedNTD G.Lee et al. (2022) | 52.31 | 58.07 | 18.03 | 97.29 | 56.43 | 0.800 | 7.90 | 10.95 | 10.89 | 14.67 | 10.34 | 11.71 | 0.911 | 0.601 | 16.77 | 18.23 | 28.47 | 93.18 | 39.16 | 0.642 | 9.932 |
| Framework for the Performance Fairness Setting § 3 | | | | | | | | | | | | | | | | | | | | | |
| AFL M.Mohri et al. (2019) | 96.58 | 90.72 | 32.90 | 87.56 | 76.94 | 0.64 | 6.57 | 85.33 | 73.79 | 80.21 | 68.93 | 77.06 | 0.775 | 0.517 | 85.76 | 72.92 | 83.16 | 87.08 | 82.23 | 0.90 | 0.329 |

10 shared categories across four domains: Amazon (Am), Caltech (Ca), DSLR (D), and Webcam (W). • **Digits** features handwritten and synthetic digit recognition across four domains: MNIST (M) (Y.LeCun et al., 1998), USPS (U) (Hull, 1994), SVHN (Svz) (Netzer et al., 2011), and SYN (Sy) (Kingma & Welling, 2013), each with ten digit classes. • **Office31** (Saenko et al., 2010) includes 31 object categories commonly seen in office environments, such as monitors, keyboards, and filing cabinets, spread across three domains: (Am, D, and W). • **PACS** comprises four stylistically varied domains: Photo (P), Art Painting (AP), Cartoon (Ct), and Sketch (Sk).

### 4.1 ADAPTATION BENCHMARK

**Evaluation Metrics.** The metric $\mathcal{A}^{\mathcal{U}}$, known as Cross-Client Accuracy, is used to evaluate performance in Cross-Client Shift scenarios, including both Label and Domain Skew settings. We further denote Out-Client Accuracy $\mathcal{A}^{\mathcal{O}}$ under Out-Client Shift for generalizable performance evaluation.

**Results:** Federated learning has been extensively explored in various settings, including Label Skew, Domain Skew, and Out-Client Shift. For the Label Skew scenario, we consider four widely used datasets: Cifar-10 (Krizhevsky et al., 2009), Cifar-100 (Krizhevsky et al., 2009), MNIST (Y.LeCun et al., 1998), and Fashion-MNIST (Xiao et al., 2017). The performance of ten methods on these datasets is summarized in Table 5. These methods range from the foundational FedAvg (McMahan et al., 2017), introduced in 2017, to more recent and sophisticated solutions (G.Lee et al., 2022). For a more detailed comparison, we also provide a visualization of the training curves, illustrating test accuracy trends during training under $\beta = 0.5$. In the case of the Domain Skew scenario, we leverage three widely used federated benchmarks: Digits (Y.LeCun et al., 1998; Hull, 1994; Y.Netzer et al., 2011; Netzer et al., 2011), Office Caltech (Fei-Fei et al., 2007; Saenko et al., 2010), and PACS. As shown in Table 6, methods like SCAFFOLD (Karimireddy et al., 2020) and FedProto (Tan et al., 2022) demonstrate relatively competitive performance across these datasets. In the Out-Client Shift setting, we evaluate Federated Domain Adaptation(FDA) and Federated Domain Generalization paradigms. FDA leverages unlabeled target distributions during training, improving *Out-Client Accuracy*. For example, KD3A achieves 67.16 accuracy on Office Caltech, demonstrating strong generalization to unseen domains.

### 4.2 TRUSTWORTHINESS BENCHMARK

**Evaluation Metrics for Robustness.** $\mathcal{A}^{u}_{Byz}$ represents the test accuracy when subjected to Byzantine Attack conditions. Consequently, the metric Accuracy Decline Impact $\mathcal{I}$ quantifies the drop in

Table 7: **Quantitative Byzantine Attack results** in term of $\mathcal{A}^u$, $\mathcal{A}^u_{Byz}$, and $\mathcal{I}$ (Appendix E.2) on Cifar-10, MNIST, and Fashion-MNIST scenarios. FLTrust and Sageflow utilizes SVHN as the proxy. The local optimization is FedProx T.Li et al. (2020a) with $\mu = 0.01$. See Byzantine Tolerance comparison in § 4.2.

| Methods | Cifar-10 | | | | | | Fashion-MNIST | | | | | | MNIST | | | | | | USPS | | | | | |
|---|---|---|---|---|---|---|---|---|---|---|---|---|---|---|---|---|---|---|---|---|---|---|---|---|
| | $\beta=0.5$ | | | $\beta=0.3$ | | | $\beta=0.5$ | | | $\beta=0.3$ | | | $\beta=0.5$ | | | $\beta=0.3$ | | | $\beta=0.5$ | | | $\beta=0.3$ | | |
| | $\Upsilon{=}0.2$ | $\Upsilon{=}0.4$ | | $\Upsilon{=}0.2$ | $\Upsilon{=}0.4$ | | $\Upsilon{=}0.2$ | $\Upsilon{=}0.4$ | | $\Upsilon{=}0.2$ | $\Upsilon{=}0.4$ | | $\Upsilon{=}0.2$ | $\Upsilon{=}0.4$ | | $\Upsilon{=}0.2$ | $\Upsilon{=}0.4$ | | $\Upsilon{=}0.2$ | $\Upsilon{=}0.4$ | | $\Upsilon{=}0.2$ | $\Upsilon{=}0.4$ | |
| | $\mathcal{A}^u_{Byz}$ | $\mathcal{A}^u_{Byz}$ | $\mathcal{I}$ | $\mathcal{A}^u_{Byz}$ | $\mathcal{A}^u_{Byz}$ | $\mathcal{I}$ | $\mathcal{A}^u_{Byz}$ | $\mathcal{A}^u_{Byz}$ | $\mathcal{I}$ | $\mathcal{A}^u_{Byz}$ | $\mathcal{A}^u_{Byz}$ | $\mathcal{I}$ | $\mathcal{A}^u_{Byz}$ | $\mathcal{A}^u_{Byz}$ | $\mathcal{I}$ | $\mathcal{A}^u_{Byz}$ | $\mathcal{A}^u_{Byz}$ | $\mathcal{I}$ | $\mathcal{A}^u_{Byz}$ | $\mathcal{A}^u_{Byz}$ | $\mathcal{I}$ | $\mathcal{A}^u_{Byz}$ | $\mathcal{A}^u_{Byz}$ | $\mathcal{I}$ |
| FedProx T.Li et al. (2020a) | $\mathcal{A}^u$:67.16 | | | :64.88 | | | $\mathcal{A}^u$:89.97 | | | :88.69 | | | $\mathcal{A}^u$:99.41 | | | :99.32 | | | $\mathcal{A}^u$:96.70 | | | :96.69 | | |
| **Pair Flipping** | | | | | | | | | | | | | | | | | | | | | | | | |
| Multi Krum Blanchard et al. (2017) | 50.21 | 46.85 | 20.31 | 46.99 | 43.91 | 20.82 | 82.20 | 47.59 | 42.38 | 80.79 | 82.51 | 6.18 | 10.18 | 11.35 | 88.06 | 10.43 | 11.35 | 87.97 | 50.83 | 93.52 | 3.18 | 93.41 | 51.11 | 45.58 |
| Bulyan R.Guerraoui et al. (2018) | 46.88 | 44.06 | 20.68 | 10.00 | 10.00 | 54.88 | 82.62 | 80.76 | 9.21 | 78.00 | 73.57 | 15.12 | 97.01 | 98.18 | 1.23 | 93.21 | 92.13 | 7.19 | 93.21 | 92.13 | 4.57 | 86.04 | 87.20 | 9.49 |
| Trim Median D.Yin et al. (2018) | 51.70 | 45.77 | 21.39 | 19.94 | 10.67 | 54.21 | 84.18 | 78.09 | 11.88 | 81.76 | 77.89 | 10.8 | 98.57 | 94.62 | 4.79 | 93.25 | 92.90 | 6.42 | 94.85 | 94.33 | 2.37 | 91.72 | 92.05 | 0.64 |
| FoolsGold Fung et al. (2018) | 60.09 | 56.80 | 10.36 | 50.81 | 57.98 | 6.90 | 86.97 | 86.07 | 3.90 | 85.65 | 81.50 | 7.19 | 97.25 | 97.80 | 1.61 | 98.05 | 97.22 | 2.10 | 77.69 | 91.77 | 4.93 | 87.90 | 77.23 | 19.46 |
| DnC Shejwalkar & Houmansadr (2021) | 62.67 | 58.38 | 8.78 | 60.41 | 59.96 | 4.92 | 87.54 | 87.76 | 2.21 | 87.22 | 88.24 | 0.45 | 99.33 | 99.07 | 0.34 | 98.85 | 98.70 | 0.62 | 95.94 | 95.16 | 1.54 | 95.07 | 95.08 | 1.61 |
| FLTrust X.Cao et al. (2021b) | / | / | / | / | / | / | / | / | / | / | / | / | 11.35 | 11.35 | 88.06 | 11.35 | 78.68 | 20.64 | 13.15 | 13.15 | 83.55 | 13.15 | 13.15 | 83.54 |
| Sageflow J.Park et al. (2021) | / | / | / | / | / | / | / | / | / | / | / | / | 99.28 | 99.03 | 0.38 | 99.02 | 98.73 | 0.59 | 95.36 | 94.34 | 2.36 | 96.15 | 95.37 | 1.32 |
| RFA K.Pillutla et al. (2022) | 66.84 | 66.31 | 0.85 | 62.28 | 61.54 | 3.34 | 89.67 | 89.73 | 0.24 | 88.18 | 88.73 | -0.04 | 99.12 | 99.10 | 0.31 | 98.97 | 98.91 | 0.41 | 96.12 | 95.56 | 1.14 | 96.30 | 96.08 | 0.61 |
| **Symmetry Flipping** | | | | | | | | | | | | | | | | | | | | | | | | |
| Multi Krum Blanchard et al. (2017) | 52.18 | 46.48 | 20.68 | 49.03 | 50.56 | 14.32 | 81.87 | 85.52 | 4.45 | 82.14 | 81.76 | 6.93 | 10.02 | 91.76 | 7.65 | 11.35 | 92.72 | 6.60 | 81.20 | 93.06 | 3.64 | 84.12 | 93.79 | 2.90 |
| Bulyan R.Guerraoui et al. (2018) | 50.73 | 38.38 | 28.78 | 14.55 | 27.01 | 37.87 | 84.15 | 82.15 | 7.82 | 79.51 | 74.93 | 13.76 | 98.50 | 97.52 | 1.89 | 87.10 | 91.66 | 7.66 | 91.46 | 89.71 | 6.99 | 89.94 | 93.70 | 2.99 |
| Trim Median D.Yin et al. (2018) | 53.24 | 49.82 | 17.34 | 34.46 | 39.24 | 25.64 | 84.61 | 84.39 | 5.58 | 80.49 | 81.48 | 7.21 | 98.50 | 98.08 | 1.33 | 92.16 | 96.25 | 3.07 | 93.46 | 92.23 | 4.47 | 93.32 | 93.70 | 2.99 |
| FoolsGold Fung et al. (2018) | 61.37 | 59.34 | 7.82 | 58.35 | 54.97 | 9.91 | 69.15 | 86.30 | 3.67 | 82.34 | 84.27 | 4.42 | 98.46 | 97.77 | 1.64 | 95.90 | 90.45 | 8.87 | 83.02 | 78.07 | 18.63 | 75.72 | 73.92 | 22.77 |
| DnC Shejwalkar & Houmansadr (2021) | 62.57 | 58.12 | 9.04 | 61.94 | 59.51 | 5.37 | 88.15 | 87.23 | 12.74 | 86.33 | 87.83 | 0.86 | 99.31 | 98.99 | 0.42 | 98.63 | 98.63 | 0.69 | 95.86 | 94.70 | 2.00 | 94.98 | 93.64 | 3.05 |
| FLTrust X.Cao et al. (2021b) | / | / | / | / | / | / | / | / | / | / | / | / | 11.35 | 70.09 | 29.32 | 11.35 | 67.29 | 32.03 | 60.41 | 52.83 | 43.87 | 59.31 | 13.15 | 83.54 |
| Sageflow J.Park et al. (2021) | / | / | / | / | / | / | / | / | / | / | / | / | 98.86 | 98.75 | 0.66 | 98.51 | 98.31 | 1.01 | 94.08 | 92.32 | 4.38 | 95.33 | 92.93 | 3.76 |
| RFA K.Pillutla et al. (2022) | 63.43 | 61.67 | 5.49 | 62.78 | 60.13 | 4.75 | 89.44 | 88.30 | 11.67 | 87.73 | 87.49 | 1.20 | 99.00 | 99.06 | 0.35 | 98.78 | 98.65 | 0.67 | 95.80 | 94.57 | 2.13 | 95.98 | 95.47 | 1.22 |
| **Random Noise** | | | | | | | | | | | | | | | | | | | | | | | | |
| Multi Krum Blanchard et al. (2017) | 10.00 | 13.06 | 54.1 | 29.25 | 14.11 | 50.77 | 10.00 | 21.71 | 68.26 | 75.55 | 25.60 | 63.09 | 11.35 | 13.42 | 85.99 | 11.35 | 21.04 | 78.28 | 89.25 | 15.07 | 81.63 | 13.15 | 26.79 | 69.90 |
| Bulyan R.Guerraoui et al. (2018) | 51.04 | 51.34 | 15.82 | 42.09 | 49.29 | 15.59 | 82.70 | 87.24 | 2.73 | 81.70 | 86.43 | 2.26 | 98.74 | 98.63 | 0.78 | 91.95 | 98.32 | 1.00 | 94.27 | 94.51 | 2.19 | 92.59 | 95.34 | 1.35 |
| Trim Median D.Yin et al. (2018) | 53.87 | 51.92 | 15.24 | 50.24 | 50.21 | 14.67 | 85.94 | 85.66 | 4.31 | 82.32 | 85.61 | 3.08 | 98.86 | 98.85 | 0.56 | 94.36 | 98.18 | 1.14 | 94.80 | 13.15 | 83.55 | 95.66 | 95.59 | 1.10 |
| FoolsGold Fung et al. (2018) | 50.01 | 32.85 | 34.31 | 49.60 | 27.45 | 37.43 | 85.98 | 35.82 | 54.15 | 76.86 | 83.58 | 5.11 | 98.46 | 37.62 | 61.79 | 87.91 | 78.90 | 20.42 | 85.36 | 22.55 | 74.15 | 54.10 | 55.92 | 40.77 |
| DnC Shejwalkar & Houmansadr (2021) | 59.64 | 56.95 | 10.21 | 60.00 | 56.45 | 8.43 | 87.81 | 87.72 | 2.25 | 87.26 | 87.66 | 1.03 | 99.31 | 98.97 | 0.44 | 98.78 | 98.85 | 0.47 | 95.73 | 94.60 | 2.10 | 95.31 | 94.28 | 2.41 |
| FLTrust X.Cao et al. (2021b) | / | / | / | / | / | / | / | / | / | / | / | / | 11.35 | 11.35 | 88.06 | 11.35 | 11.35 | 87.97 | 36.53 | 13.15 | 83.55 | 13.15 | 13.15 | 83.54 |
| Sageflow J.Park et al. (2021) | / | / | / | / | / | / | / | / | / | / | / | / | 98.76 | 96.75 | 2.66 | 93.14 | 89.85 | 9.47 | 92.40 | 78.20 | 18.50 | 86.02 | 75.63 | 21.06 |
| RFA K.Pillutla et al. (2022) | 56.37 | 10.64 | 56.52 | 55.88 | 15.45 | 49.43 | 87.11 | 64.10 | 25.87 | 85.32 | 72.30 | 16.39 | 99.15 | 95.40 | 4.01 | 98.26 | 94.01 | 5.31 | 94.67 | 67.49 | 29.21 | 95.35 | 53.08 | 43.61 |
| **Min-Sum** | | | | | | | | | | | | | | | | | | | | | | | | |
| Multi Krum Blanchard et al. (2017) | 10.00 | 10.90 | 56.26 | 42.20 | 10.02 | 54.86 | 10.00 | 11.02 | 78.95 | 80.78 | 10.00 | 78.69 | 11.35 | 23.17 | 76.24 | 10.43 | 11.35 | 87.97 | 13.15 | 15.96 | 80.74 | 13.15 | 13.15 | 83.54 |
| Bulyan R.Guerraoui et al. (2018) | 51.49 | 51.00 | 16.16 | 42.99 | 40.07 | 24.81 | 84.64 | 85.84 | 4.13 | 80.23 | 84.21 | 4.48 | 98.60 | 94.38 | 5.03 | 92.40 | 90.14 | 9.18 | 94.88 | 85.91 | 10.79 | 92.91 | 93.36 | 3.33 |
| Trim Median D.Yin et al. (2018) | 53.62 | 53.71 | 13.45 | 48.58 | 51.76 | 13.12 | 84.64 | 85.71 | 4.26 | 83.24 | 85.41 | 3.28 | 98.77 | 98.76 | 0.65 | 96.80 | 92.90 | 6.42 | 95.12 | 95.75 | 0.95 | 94.22 | | |
| FoolsGold Fung et al. (2018) | 52.26 | 10.00 | 57.16 | 47.83 | 10.00 | 54.88 | 80.58 | 14.80 | 75.17 | 80.20 | 19.36 | 69.33 | 97.18 | 16.87 | 82.54 | 98.71 | 97.22 | 2.10 | 69.49 | 15.04 | 81.66 | 64.16 | 13.12 | 83.57 |
| DnC Shejwalkar & Houmansadr (2021) | 61.11 | 55.52 | 11.84 | 60.29 | 55.83 | 9.05 | 87.63 | 87.80 | 2.17 | 87.25 | 88.01 | 0.68 | 99.20 | 99.20 | 0.21 | 98.80 | 98.70 | 0.62 | 95.34 | 94.51 | 2.19 | 94.93 | 95.35 | 1.34 |
| FLTrust X.Cao et al. (2021b) | / | / | / | / | / | / | / | / | / | / | / | / | 61.57 | 12.99 | 86.42 | 11.35 | 11.35 | 87.97 | 13.15 | 15.04 | 81.66 | 13.15 | 14.09 | 82.60 |
| Sageflow J.Park et al. (2021) | / | / | / | / | / | / | / | / | / | / | / | / | 98.59 | 92.85 | 6.56 | 92.30 | 85.01 | 14.31 | 87.07 | 14.09 | 82.61 | 81.95 | 50.59 | 46.1 |
| RFA K.Pillutla et al. (2022) | 51.90 | 11.40 | 55.76 | 60.29 | 14.22 | 50.66 | 87.40 | 22.83 | 67.14 | 85.71 | 61.18 | 27.51 | 99.05 | 94.39 | 5.02 | 98.80 | 98.91 | 0.41 | 94.65 | 71.23 | 25.47 | 94.93 | 57.83 | 38.86 |

performance relative to standard (benign) federated learning. Likewise, Attack Success Rate $\mathcal{R}^u$ measures model behavior on datasets affected by backdoor attacks.

**Results:** Table 7 summarizes the experimental outcomes for various Byzantine Tolerance strategies under Byzantine Attack scenarios. The evaluation is conducted on four widely used datasets: Cifar-10, Fashion-MNIST, MNIST, and USPS. We examine two categories of data poisoning attacks, specifically Data-Based Byzantine Attack techniques: Pair Flipping and Symmetry Flipping. Additionally, we investigate two model poisoning approaches under Model-Based Byzantine Attack, namely Random Noise and Min-Sum. The selected Byzantine Tolerance approaches fall into three categories: Distance Base Tolerance, Statistics Distribution Tolerance, and Proxy Dataset Tolerance. Among them, DnC demonstrates comparatively strong resilience across all attack types. In contrast, methods under the Proxy Dataset Tolerance category exhibit notable limitations, often requiring external proxy data. Table 8 presents the results for Backdoor Attack namely two prevalent variants: Bac and Sem Bac. Additionally, we assess the robustness of two prominent Backdoor Defense techniques, namely RLR (Ozdayi et al., 2021) and CRFL (C.Xie et al., 2021), having effective defense capabilities against backdoor threats.

**Evaluation Metrics for fairness.** As described in § A.2.1, Contribution Match Degree ($\mathcal{E}$) and Performance Deviation ($\mathcal{V}$) are metrics specifically designed to assess Performance Fairness..

**Results:** As shown in Table 5 and Table 6, few of the existing federated optimization takes the Collaboration Fairness into federated objective account. Besides, fairness is also largely impeded under large local data distribution diversity, such as the Domain Skew. Regarding the Performance Fairness, existing methods focus on minimizing the weighted empirical loss and thus bring the imbalanced performance. Notably, global network utilization and server adaptive optimization seem to alleviate the imbalanced performance on the multiple domains roundly.

## 5 FUTURE OUTLOOK

**(1) Summary of Experimental Observations.** Our evaluation surfaces key trends and gaps across federated learning methods: • *Reproducibility Dilemma.* Many FL studies lack transparent ex-

Table 8: **Quantitative Backdoor Attack results** in term of $\mathcal{A}^u$ and $\mathcal{R}^u$ on Cifar-10, MNIST, and USPS. The local optimization algorithm is FedAvg McMahan et al. (2017). We consider two types of backdoor attacks and abbreviate them as Bac X.Chen et al. (2017) and Sem Bac E.Bagdasaryan et al. (2020). - means that these solutions are not applicable to these evaluations. Refer to § 4.2 for Backdoor Defense discussion.

| Methods | Cifar-10 | | | | | | | | MNIST | | | | | | | | USPS | | | | | | | |
|---|---|---|---|---|---|---|---|---|---|---|---|---|---|---|---|---|---|---|---|---|---|---|---|---|
| | 0.5 | | | | 0.3 | | | | 0.5 | | | | 0.3 | | | | 0.5 | | | | 0.3 | | | |
| | Bac | | Sem Bac | | Bac | | Sem Bac | | Bac | | Sem Bac | | Bac | | Sem Bac | | Bac | | Sem Bac | | Bac | | Sem Bac | |
| | $\mathcal{A}^u$ | $\mathcal{R}^u$ | $\mathcal{A}^u$ | $\mathcal{R}^u$ | $\mathcal{A}^u$ | $\mathcal{R}^u$ | $\mathcal{A}^u$ | $\mathcal{R}^u$ | $\mathcal{A}^u$ | $\mathcal{R}^u$ | $\mathcal{A}^u$ | $\mathcal{R}^u$ | $\mathcal{A}^u$ | $\mathcal{R}^u$ | $\mathcal{A}^u$ | $\mathcal{R}^u$ | $\mathcal{A}^u$ | $\mathcal{R}^u$ | $\mathcal{A}^u$ | $\mathcal{R}^u$ | $\mathcal{A}^u$ | $\mathcal{R}^u$ | $\mathcal{A}^u$ | $\mathcal{R}^u$ |
| *Focus on Byzantine Tolerance § 3* | | | | | | | | | | | | | | | | | | | | | | | | |
| Bulyan R.Guerraoui et al. (2018) | 47.61 | 28.73 | 44.61 | 17.12 | - | - | 11.12 | 19.56 | 96.95 | 14.77 | 92.13 | 0.45 | 87.70 | 11.13 | **87.86** | **0.10** | 93.32 | 10.95 | 93.52 | 11.32 | 87.79 | 10.83 | 85.14 | 1.56 |
| Trim Median D.Yin et al. (2018) | 51.34 | **22.49** | 52.21 | 13.70 | - | - | 14.78 | 51.66 | 98.07 | 99.18 | 98.44 | 0.16 | 96.65 | 89.42 | 96.72 | 0.61 | 94.62 | 71.52 | 94.24 | 4.82 | 92.05 | 84.17 | 94.77 | 2.40 |
| FoolsGold Fung et al. (2018) | **60.69** | 62.54 | 60.50 | 13.06 | 58.58 | 56.85 | 59.84 | 12.56 | 82.20 | 91.61 | 98.45 | 0.59 | 92.88 | 98.06 | 97.00 | 1.52 | 89.66 | 90.24 | 83.21 | 10.11 | 76.56 | 86.14 | 94.77 | 2.40 |
| DnC Shejwalkar & Houmansadr (2021) | 59.30 | 23.07 | **61.40** | 12.88 | 60.03 | **42.79** | 59.80 | 9.76 | **99.26** | 10.39 | 99.13 | 0.20 | 98.53 | 10.46 | 98.79 | 0.29 | 95.75 | 9.62 | **95.11** | 2.89 | 96.14 | 16.89 | 94.86 | 1.81 |
| FLTrust X.Cao et al. (2021b) | / | / | / | / | / | / | / | / | 95.31 | **8.71** | 97.84 | **0.00** | 92.55 | **10.03** | 97.43 | 0.30 | 71.67 | 17.69 | 59.83 | 20.96 | **63.20** | **5.29** | 63.20 | 5.29 |
| Sageflow J.Park et al. (2021) | / | / | / | / | / | / | / | / | 99.17 | 98.70 | **99.21** | 0.53 | 99.03 | 98.05 | 98.83 | 1.27 | 96.07 | 73.63 | 96.20 | 3.61 | 96.83 | 86.39 | 96.02 | 2.65 |
| RFA K.Pillutla et al. (2022) | **64.90** | 74.31 | **63.90** | 11.54 | **60.36** | 75.57 | **62.75** | 14.76 | 99.09 | 99.09 | 99.12 | 0.32 | **99.11** | 98.88 | 98.84 | 0.39 | **95.89** | **2.28** | 95.75 | 3.13 | 97.04 | 39.59 | 95.89 | 2.28 |
| *Focus on Backdoor Defense* | | | | | | | | | | | | | | | | | | | | | | | | |
| RLR Ozdayi et al. (2021) | 51.65 | 28.83 | 50.37 | 10.60 | - | - | 44.80 | 20.74 | 94.77 | 10.54 | 93.11 | 0.40 | 91.11 | 22.69 | 92.94 | 0.35 | 89.20 | 10.78 | 92.00 | 12.65 | 87.00 | 10.27 | 82.15 | 1.44 |
| CRFL C.Xie et al. (2021) | 59.27 | 63.29 | 58.59 | **9.52** | 52.27 | 59.50 | 52.62 | 11.66 | 98.93 | 33.86 | 98.89 | 0.43 | 98.44 | 26.28 | 98.08 | 0.91 | 94.96 | 49.77 | 95.31 | 3.61 | 95.38 | 62.98 | **94.36** | **1.32** |

perimental setups and open-source code. The inconsistency in datasets and models complicates fair comparisons, undermining reproducibility. ● *Computational Efficiency Gap.* Despite strong accuracy claims, most methods overlook memory and runtime overheads. In real-world deployments, especially cross-device (Hard et al., 2018) and cross-silo (Yoo et al., 2021; Yang et al., 2019) settings, efficiency is often a limiting factor. ● *Fragmented Solutions.* FL research often targets isolated issues like heterogeneity (X.Ma et al., 2022), robustness (J.Shi et al., 2022), or fairness (Y.Shi et al., 2023a), lacking unified solutions that balance performance, trust, and efficiency.

**(2) Open Issues and Future Opportunities.** ● *Building a Reasoning Benchmark.* Our work highlights reasoning as a critical next frontier for FL evaluation. Future efforts should focus on establishing dedicated benchmarks and defining evaluation criteria for trace coherence, faithfulness, and privacy-preserving reasoning across decentralized clients. ● *Towards Reproducibility.* **FLAT-Bench** introduces a unified taxonomy, standard protocols, and open-source assets to enhance comparability. Future work should prioritize consistent baselines and transparent reporting practices. ● *Advancing Efficiency.* While optimizations like quantization, pruning, and homomorphic encryption (Shao et al., 2024) have emerged, trade-offs remain. Future FL systems must balance speed, scalability, and security to support edge-centric applications. ● *Toward Holistic Evaluation.* We advocate for comprehensive benchmarks that jointly assess generalization, robustness, fairness, reasoning, and efficiency across diverse modalities including video and multimodal settings to close the gap between research and deployment.

# 6 CONCLUSION

We present **FLAT-Bench**, the first comprehensive benchmark designed to systematically evaluate federated learning (FL) across two foundational pillars: *Adaptation* and *Trust*. Our benchmark organizes a broad range of FL methods by task settings, learning strategies, and their respective contributions, offering a structured lens through which to assess progress in the field. Through extensive empirical evaluation across eight widely used FL datasets, FLAT-Bench reveals key trends, challenges, and performance bottlenecks, shedding light on critical areas for improvement. By surfacing these insights, FLAT-Bench lays a solid foundation for the development of more robust, trustworthy, and adaptable federated learning systems, ultimately supporting both future research and real-world deployment.

# 7 LIMITATIONS

Despite its contributions, **FLAT-Bench** has limitations. Benchmarking reasoning capabilities in large language models (LLMs) remains an open challenge, particularly in federated settings where reasoning trajectories can vary significantly across clients. Our benchmark underscores this gap and highlights the urgent need for unified, standardized metrics to evaluate the coherence, faithfulness, and adaptability of distributed reasoning. Addressing this limitation is essential for advancing trustworthy FL systems, especially in domains that demand transparent and interpretable model behavior.

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

APPENDIX

# A  BACKGROUND

## A.1  HISTORY AND TERMINOLOGY

Federated learning enables multiple parties to jointly train a shared model without exchanging their raw data, preserving privacy and reducing communication overhead. Early formulations include client-server optimization schemes and federated averaging algorithms (Konečnỳ et al., 2016a;b; McMahan et al., 2017). Depending on how data are partitioned across participants, FL methods are typically divided into three paradigms (H.Zhu et al., 2021; Rodríguez-Barroso et al., 2023):

- **Horizontal Federated Learning (HFL)**: All clients hold data with the same feature space but on different samples. They collaboratively update a global model by sharing parameter updates while keeping each local dataset private (McMahan et al., 2017; Miao et al., 2023).
- **Vertical Federated Learning (VFL)**: Participants possess complementary features for the same set of entities. Secure protocols are used to jointly compute model updates on aligned samples without revealing individual feature values (Liu et al., 2022; Wei et al., 2022).
- **Federated Transfer Learning (FTL)**: When both feature spaces and sample sets differ across clients, FTL applies transfer learning techniques—such as knowledge distillation or representation mapping—to enable knowledge sharing between heterogeneous domains (Liu et al., 2020b; Saha & Ahmad, 2021).

In this work, we concentrate on four fundamental properties of horizontal federated learning (HFL)[1] and present a unified evaluation framework under the HFL setting: Generalization (GFL). Due to the non-IID nature of client data, federated models must contend with both cross-client distribution shifts—where local empirical risks diverge across participants—and out-of-client distribution shifts, which capture discrepancies between seen and unseen client populations (Li et al., 2019; X.Peng et al., 2020; Q.Liu et al., 2021). These phenomena hinder both convergence speed and test-time performance when models are deployed on new or held-out clients. Robustness (RFL). Federated learning's decentralized paradigm exposes it to adversarial manipulation. On one hand, Byzantine attacks corrupt either local training data or uploaded updates to derail global aggregation (L.Huang et al., 2011; Damaskinos et al., 2018). On the other, backdoor attacks stealthily inject triggers into client updates so that the global model behaves normally on benign inputs but misclassifies targeted samples (Sun et al., 2019; E.Bagdasaryan et al., 2020). Fairness (FFL). Equitable participation and performance are critical to sustain federated collaborations. Collaborative fairness addresses how to reward clients proportionally to their computational effort and data value (T.Song et al., 2019; Nguyen et al., 2022b), while performance fairness ensures that the global model does not systematically underperform on underrepresented or marginalized client distributions (M.Mohri et al., 2019; Cui et al., 2021). By benchmarking these two axes: generalization and robustness under a common HFL protocol, we aim to provide a comprehensive assessment of federated methods and elucidate their trade-offs for real-world, privacy-sensitive deployments.

## A.2  PROBLEM FORMULATION

We consider a horizontal federated learning setting with $M$ clients, indexed by $i = 1, \ldots, M$, each holding a private dataset $\mathcal{D}_i$ of size $N_i = |\mathcal{D}_i|$. Each example $(x, y) \in \mathcal{D}_i$ is drawn from a client-specific distribution $\mathbb{P}_i(x, y)$. Our goal is to train a shared model

$$w = f \circ g,$$

where $f : \mathcal{X} \to \mathbb{R}^d$ is a feature extractor mapping inputs $x$ to $d$-dimensional embeddings $h = f(x)$, and $g : \mathbb{R}^d \to \mathbb{R}^{|C|}$ is a classifier producing logits $z = g(h)$ over the label set $C$.

Federated learning seeks the global parameter $w^*$ that minimizes a weighted combination of local empirical risks:

$$w^* = \arg\min_w \sum_{i=1}^{M} \alpha_i \, \mathcal{L}_i(w; \mathcal{D}_i), \tag{1}$$

---

[1]We use "HFL" to denote horizontal federated learning.

where $\mathcal{L}_i(w; \mathcal{D}_i) = \frac{1}{N_i} \sum_{(x,y) \in \mathcal{D}_i} \ell\big(g(f(x)), y\big)$ is the average loss on client $i$, and the mixing weights satisfy $\sum_i \alpha_i = 1$ (commonly $\alpha_i = N_i / \sum_j N_j$ or $\alpha_i = 1/M$).

Training proceeds in communication rounds, each consisting of three phases:

$$
\begin{aligned}
&\textbf{1. Broadcast:} && w_i^{(t)} = w^{(t-1)} \quad \forall i, \\
&\textbf{2. Local Update:} && w_i^{(t)} \leftarrow \arg\min_{w_i} \mathbb{E}_{(x,y) \sim \mathcal{D}_i}\big[\ell\big(g(f(x; w_i)), y\big)\big], \\
&\textbf{3. Aggregation:} && w^{(t)} = \sum_{i=1}^{M} \alpha_i \, w_i^{(t)}.
\end{aligned}
\tag{2}
$$

Here, step 1 distributes the current global model to all clients; step 2 performs one or more epochs of local optimization (e.g. via SGD) on each $\mathcal{D}_i$; and step 3 fuses client updates into the new global model. This iterative protocol continues until convergence or a stopping criterion is met (McMahan et al., 2017; T.Li et al., 2020a)."'

### A.2.1 DATA HETEROGENEITY IN FEDERATED LEARNING

In real-world federated setups, each client's dataset $\mathcal{D}_i$ is drawn from its own distribution $\mathbb{P}_i(x, y)$, leading to non-IID data across the network (T.Li et al., 2020a; Q.Liu et al., 2021; Qu et al., 2021). We often decompose $\mathbb{P}_i(x, y) = \mathbb{P}_i(y)\,\mathbb{P}_i(x \mid y)$ and distinguish two principal forms of heterogeneity:

- **Label shift:** Clients differ in their label marginals but share the same class-conditional features:
$$\mathbb{P}_i(y) \neq \mathbb{P}_j(y), \quad \mathbb{P}_i(x \mid y) = \mathbb{P}_j(x \mid y).$$
A common simulation uses Dirichlet sampling (Kotz et al., 2004) to skew $\mathbb{P}_i(y)$.

- **Feature shift:** All clients have the same label distribution but observe different feature patterns for each class:
$$\mathbb{P}_i(y) = \mathbb{P}_j(y), \quad \mathbb{P}_i(x \mid y) \neq \mathbb{P}_j(x \mid y).$$
This arises, for example, when imaging devices vary across hospitals (X.Li et al., 2021).

Beyond these in-network shifts, **out-of-client shift** refers to the performance degradation when deploying the federated model on entirely new data sources $\mathbb{P}_o(x, y) \neq \mathbb{P}_i(x, y)$, despite matching label marginals:
$$\mathbb{P}_o(y) = \mathbb{P}_i(y), \quad \mathbb{P}_o(x \mid y) \neq \mathbb{P}_i(x \mid y).$$
Such unseen domain shifts underscore the need for federated methods that generalize beyond the participating clients (H.Yuan et al., 2022).

### A.2.2 ADVERSARIAL THREATS IN FEDERATED LEARNING

In federated settings, untrusted participants may launch attacks that compromise model integrity. We categorize these into two broad classes:

**1. Byzantine (Untargeted) Attacks** Here, adversaries aim to simply degrade overall model accuracy without a specific target outcome (Blanchard et al., 2017; R.Guerraoui et al., 2018; Damaskinos et al., 2018). Two common strategies are:

- **Data Poisoning:** Malicious clients corrupt their local training data before participating. For example, in symmetric label noise (SymFlip), each label is flipped to any other class with equal probability $\epsilon/(|C|-1)$:
$$
T_{\text{sym}}(i, j) = \begin{cases} 1 - \epsilon & i = j, \\ \frac{\epsilon}{|C|-1} & i \neq j, \end{cases}
$$
while in pair-flip noise (PairFlip) labels are only swapped among semantically similar classes (B.VanRooyen et al., 2015; B.Han et al., 2018).

- **Model Poisoning:** Rather than tampering with data, adversaries directly alter their client updates. Examples include:
  - *Random-Noise:* Substituting the true gradient $\nabla_k$ with random values (e.g., Gaussian noise).
  - *Lie Attack:* Crafting updates just beyond detection thresholds by adding a small multiple of the benign update standard deviation (G.Baruch et al., 2019).
  - *Optimization-Aware Poisoning:* Solving a max-loss subproblem to push the global model away from its benign update trajectory (M.Fang et al., 2020).
  - *MinMax/MinSum Attacks:* Adjusting the poisoned update so that its maximum (or sum) distance to benign updates remains within the natural benign update spread (Shejwalkar & Houmansadr, 2021).

**2. Backdoor (Targeted) Attacks**  Here, the attacker embeds a hidden trigger so that when specific patterns are present, the global model misclassifies inputs into a chosen target label, while preserving normal performance otherwise (X.Chen et al., 2017; C.Liao et al., 2018). Concretely, poisoned clients mix a trigger mask $m$ and pattern $\Phi$ into a fraction of their examples:

$$\widetilde{x} = (1 - m) \odot x + m \odot \Phi,$$

and optimize a combined loss:

$$\mathbb{E}_{(x,y)\sim D_i}\big[L(w_i, x, y)\big] \ + \ \lambda\, \mathbb{E}_{(\widetilde{x}, y_t)}\big[L(w_i, \widetilde{x}, y_t)\big],$$

where $y_t$ is the attacker-specified target class and $\lambda \geq 0$ balances backdoor potency against clean-data fidelity. Recent work has shown that distributing trigger fragments across multiple malicious clients can evade standard defenses (C.Xie et al., 2020a; X.Lyu et al., 2023).

### A.2.3   CLIENT INCENTIVES AND FAIRNESS

Federated learning relies on voluntary participation of clients with heterogeneous data and compute resources. To maintain long-term engagement and equitable outcomes, two primary fairness concerns must be addressed:

**Reward Allocation (Reward Conflict)**  Clients incur varying costs (e.g., data labeling, computation) and contribute unequally to the global model's performance (X.Zhang et al., 2020; Y.Shi et al., 2023a). A fair compensation scheme should grant higher rewards to those whose participation yields larger marginal gains. We adopt the Shapley Value from cooperative game theory (Shapley, 1997; Bilbao, 2012; M.Davis & M.Maschler, 1965) to quantify each client's contribution:

$$\nu_i \ = \ \frac{\rho}{M} \sum_{S \subseteq \{1,...,M\}\setminus\{i\}} \frac{A\big(w_{S\cup\{i\}}, u\big) - A\big(w_S, u\big)}{\binom{M-1}{|S|}},$$

where $A(w_S, u)$ is the model accuracy on test set $u$ when trained on clients in $S$, and $\rho > 0$ scales the values.

**Prediction Consistency (Prediction Biases)**  Data heterogeneity can cause the global model to perform well on some client domains but poorly on others, leading to prediction bias (M.Mohri et al., 2019; T.Li et al., 2020b). We measure this by the standard deviation of per-domain accuracies:

$$\zeta \ = \ \mathrm{StdDev}\big(\{A(w, u)\}_{u \in \mathcal{U}}\big),$$

where $\mathcal{U}$ is the set of evaluation domains. Lower $\zeta$ indicates more uniform performance, while higher $\zeta$ signals greater disparity among client groups.

## B   HYPERPARAMETERS

## C   RELATED WORK

Federated learning (FL) has spawned numerous survey papers in recent years. Early overviews (Yang et al., 2019; T.Li et al., 2020c; Wahab et al., 2021; Q.Li et al., 2021b; P.Kairouz et al., 2021; Rodríguez-Barroso et al., 2023) lay out the high-level principles and system challenges, but typically do not

Table 9: Selected hyper-parameters for the various evaluated methods. Note that similar symbols may represent **different concepts** across different approaches. Detailed explanations are provided in Appendix F.2.

| Method | Hyper-Parameters |
|---|---|
| **General FL Methods (Generalizable Federated Learning)** .. § 2 | |
| FedProx T.Li et al. (2020a) | Proximal term $\mu = 0.01$ |
| SCAFFOLD Karimireddy et al. (2020) | Server-side learning rate $lr = 0.25$ |
| FedProc X.Mu et al. (2021) | Contrastive temperature $\tau = 1.0$ |
| MOON Q.Li et al. (2021a) | $\tau = 0.5$ (temp), $\mu = 1.0$ (proximal) |
| FedRS Li & Zhan (2021) | Scaling factor $\alpha = 0.5$ |
| FedDyn X.Mu et al. (2021) | Regularization strength $\alpha = 0.5$ |
| FedOpt Reddi et al. (2021) | Global optimizer LR $\eta_g = 0.5$ |
| FedProto Tan et al. (2022) | Prototype regularizer $\lambda = 2$ |
| FedLC Zhang et al. (2022) | Scaling factor $\tau = 0.5$ |
| FedDC L.Gao et al. (2022) | Penalty weight $\alpha = 0.1$ |
| FedNTD G.Lee et al. (2022) | Temp $\tau = 1$, Reg weight $\beta = 1$ |
| FPL Huang et al. (2023) | Contrastive temperature $\tau = 0.02$ |
| KD3A H.Feng et al. (2021) | Confidence gate $g \in [0.9, 0.95]$ |
| **Robust FL Methods (Robust Federated Learning)** ..... § 3 | |
| Multi-Krum Blanchard et al. (2017) | Byzantine tolerance $\Upsilon < 50\%$, Top-K: 5 |
| Bulyan R.Guerraoui et al. (2018) | Byzantine tolerance $\Upsilon < 50\%$ |
| Trimmed Mean D.Yin et al. (2018) | Evil client ratio $\Upsilon < 50\%$ |
| FoolsGold Fung et al. (2018) | Stability threshold $\epsilon = 10^{-5}$ |
| DnC Shejwalkar & Houmansadr (2021) | Sub-dim $b = 1000$, filter ratio $c = 1.0$ |
| FLTrust X.Cao et al. (2021b) | Public epochs $E = 20$ |
| SageFlow J.Park et al. (2021) | Threshold $E_{th} = 2.2$, exponent $\delta = 5$ |
| RFA K.Pillutla et al. (2022) | Iterations $E = 3$ |
| RLR Ozdayi et al. (2021) | LR $lr = 1.0$, threshold $\tau = 4.0$ |
| CRFL C.Xie et al. (2021) | Norm threshold $\rho = 15$, smoothing $\sigma = 0.01$ |
| **Fairness-Oriented FL Methods (Fair Federated Learning)** . § 3 | |
| AFL M.Mohri et al. (2019) | Regularization coefficient $\gamma = 0.01$ |

delve into detailed algorithmic solutions for specific FL problems. A large body of work addresses distributional heterogeneity in FL. Several surveys (Y.Zhao, 2018; H.Zhu et al., 2021; Q.Li et al., 2022; M.Ye et al., 2023; Y.Li et al., 2023) categorize approaches for label skew, feature skew, and concept drift between clients, and compare client-level strategies such as local regularization (T.Li et al., 2020a), personalized layers (Liu et al., 2024a), and meta-learning (Fallah et al., 2020). Domain adaptation in FL—where some target domain data are available during training—is surveyed in (X.Peng et al., 2020; H.Yuan et al., 2022), highlighting adversarial alignment (G.Li et al., 2023) and feature disentanglement (G.Wu & S.Gong, 2021). Out-of-distribution generalization methods, which aim to perform well on unseen client distributions, are comparatively less reviewed but include invariant optimization (Q.Liu et al., 2021) and robust aggregation schemes (Duan et al., 2023). FL's distributed nature makes it vulnerable to Byzantine and backdoor attacks. Surveys on adversarial threats (L.Lyu et al., 2020a; J.Shi et al., 2022; J.Shao et al., 2023) classify untargeted data and model

poisoning (e.g., (Blanchard et al., 2017; R.Guerraoui et al., 2018)) and targeted backdoors (Sun et al., 2019; E.Bagdasaryan et al., 2020). Defense surveys (V.Mothukuri et al., 2021) compare robust aggregation, anomaly detection, and certified defenses (X.Cao et al., 2021a; C.Xie et al., 2021). Fairness in FL encompasses both equitable performance across client groups and fair reward allocation. Recent reviews (Rafi et al., 2023; Y.Shi et al., 2023a) discuss methods that enforce uniform accuracy via min–max optimization (M.Mohri et al., 2019; T.Li et al., 2020b) or multi-objective updates (Z.Hu et al., 2020). Client-level incentive mechanisms based on reputations (L.Lyu et al., 2020c) and data valuation via Shapley approximations (X.Xu et al., 2021; Jiang et al., 2023) are surveyed in (Q.Li et al., 2021b). As FL moves into high-stakes domains, model transparency and reasoning become critical. While most surveys focus on performance, a few emerging works (Liu et al., 2023a) explore integrating chain-of-thought explanations into FL, and others (Song et al., 2021) survey symbolic and knowledge-graph based federated models. However, there is no comprehensive survey that brings together domain adaptation, generalization, robustness, fairness, and reasoning under a unified evaluation framework. To fill these gaps, we present the first holistic survey and benchmark that jointly examines *domain adaptation*, *OOD generalization*, *adversarial robustness*, *fairness*, and *reasoning* in FL. We systematically categorize state-of-the-art methods in each dimension and provide a unified empirical comparison across common benchmarks, offering both breadth and depth for researchers and practitioners.

## D OUTLINE

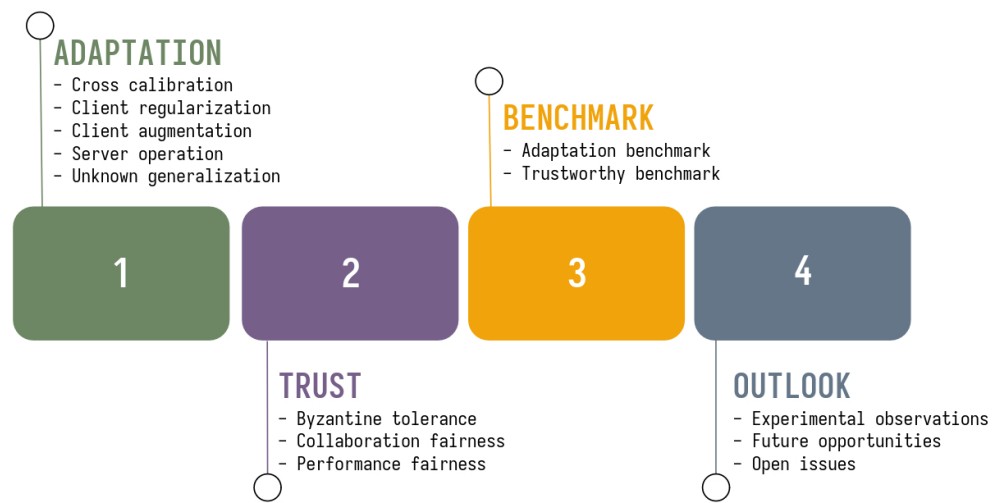

FLAT-Bench framework is organized around four key components, each addressing a foundational aspect of federated learning. Adaptation focuses on techniques that enhance generalization across diverse clients, including regularization, augmentation, and cross-domain calibration. Trust centers on robustness in adversarial and unreliable environments, covering Byzantine resilience and fairness across both collaboration and performance. The Benchmark module formalizes these dimensions through standardized evaluations, enabling consistent comparisons across methods and datasets. Finally, Outlook offers reflective insights, summarizing experimental findings and outlining future research opportunities. Together, these pillars form a structured foundation for evaluating, comparing, and advancing federated learning in real-world settings.

# E    BENCHMARK METRICS

## E.1    GENERALIZATION METRICS

We evaluate a federated model's ability to handle distribution shifts in two scenarios: *cross-client* and *out-of-distribution*.

**Cross-Client Accuracy.**    Under cross-client heterogeneity, each client's test set $u$ may follow a different distribution. We measure the standard Top-1 accuracy on each $u$ as

$$A_u = \frac{1}{|u|} \sum_{(x,y) \in u} \mathbf{1}\{\arg\max w(x) = y\},$$

and report the mean over a collection of held-out client sets $\mathcal{U}$ via

$$A_{\mathcal{U}} = \frac{1}{|\mathcal{U}|} \sum_{u \in \mathcal{U}} A_u.$$

Results across held-out clients under various distribution shifts are summarized in Table 10.

**Out-of-Distribution Accuracy.**    To assess performance on entirely unseen domains, we compute Top-1 accuracy on a designated OOD test set $O$:

$$A_O = \frac{1}{|O|} \sum_{(x,y) \in O} \mathbf{1}\{\arg\max w(x) = y\}.$$

## E.2    ROBUSTNESS METRICS

In federated learning, adversarial participants can undermine the shared model through untargeted (Byzantine) or targeted (backdoor) manipulations. We quantify defense effectiveness with two key metrics:

**Accuracy Degradation ($I$).**    For Byzantine resilience, compare the model's clean accuracy $A_{\text{clean}}$ on domain $u$ against its accuracy under attack $A_{\text{byz}}$. The degradation

$$I = A_{\text{clean}} - A_{\text{byz}}$$

measures how much performance is lost due to malicious updates.

**Backdoor Success Rate ($R$).**    To assess backdoor defenses, we inject a trigger into each test sample, yielding $(\tilde{x}, \tilde{y})$, and record the fraction that the global model misclassifies as the attacker's target label:

$$R = \frac{1}{|\tilde{T}|} \sum_{(\tilde{x}, \tilde{y}) \in \tilde{T}} \mathbf{1}\{\arg\max w(\tilde{x}) = \tilde{y}\},$$

where $\tilde{T}$ is the set of all poisoned examples.

## E.3    FAIRNESS METRICS

In federated learning, participants incur varying costs and offer data of unequal value, making fair reward allocation and uniform performance critical. The federated settings we experiment with are outlined in Table 11. We capture these with two complementary metrics:

**Contribution Impact ($\mathcal{C}$).**    Rather than using static weights $\alpha_i$, we quantify each client's real influence on global accuracy by a leave-one-out procedure. Let $w$ be the assembled global model and $w_i$ the contribution from client $i$. Excluding $i$ yields

$$w^{-i} = \frac{w - \alpha_i \, w_i}{1 - \alpha_i} \, .$$

Table 10: **Performance under Out-Client Shift** setting, reported using the metric $\mathcal{A}^O$, across the Office Caltech, Digits, PACS, and Office31 benchmarks. Refer to § 4.1 for detailed analysis.

| Methods | Office Caltech | | | | | Digits | | | | | PACS | | | | | Office31 | | | |
|---|---|---|---|---|---|---|---|---|---|---|---|---|---|---|---|---|---|---|---|
| | →Ca | →Am | →W | →D | AVG | →M | →U | →Svz | →Sy | AVG | →P | →AP | →Ct | →Sk | AVG | →D | →Am | →W | AVG |
| FedAvg McMahan et al. (2017) | 58.12 | 67.47 | 43.10 | 80.00 | 62.17 | 32.60 | 47.20 | 13.91 | 13.54 | 26.81 | 52.28 | 46.16 | 60.74 | 51.12 | 52.57 | 14.28 | 8.93 | 21.51 | 14.90 |
| FedProx T.Li et al. (2020a) | 56.60 | 69.26 | 42.41 | 85.33 | 63.40 | 23.54 | 60.28 | 15.83 | 13.78 | 28.35 | 54.45 | 49.61 | 56.91 | 56.17 | 54.28 | 15.92 | 6.01 | 19.36 | 13.76 |
| SCAFFOLD Karimireddy et al. (2020) | 36.07 | 47.36 | 45.86 | 59.33 | 47.15 | 67.61 | 82.39 | 7.79 | 14.52 | **43.07** | 43.85 | 23.81 | 45.07 | 39.79 | 38.12 | 12.44 | 5.58 | 10.88 | 9.63 |
| FedProc X.Mu et al. (2021) | 47.41 | 60.84 | 42.41 | 66.66 | 54.33 | 24.34 | 43.37 | 10.15 | 13.09 | 22.73 | 56.94 | 30.95 | 56.02 | 49.94 | 48.46 | 19.39 | 4.91 | 10.38 | 11.56 |
| MOON Q.Li et al. (2021a) | 55.53 | 68.63 | 44.83 | 79.33 | 62.08 | 31.28 | 31.75 | 14.30 | 14.45 | 22.94 | 54.01 | 45.10 | 60.42 | 58.10 | 54.40 | 14.08 | 7.04 | 21.39 | 14.17 |
| FedDyn Acar et al. (2021) | 59.99 | 66.42 | 40.34 | 81.99 | 62.18 | 28.74 | 56.08 | 14.36 | 11.88 | 27.76 | 51.40 | 43.19 | 60.57 | 50.71 | 51.46 | 14.08 | 7.86 | 17.85 | 13.26 |
| FedOPT Reddi et al. (2021) | 52.67 | 55.68 | 60.34 | 69.33 | 59.50 | 59.35 | 62.62 | 17.59 | 15.22 | 38.69 | 57.64 | 39.19 | 45.92 | 49.50 | 48.06 | 19.38 | 6.90 | 18.73 | 15.00 |
| FedProto Tan et al. (2022) | 60.35 | 66.94 | 58.62 | 76.00 | **65.47** | 43.67 | 58.08 | 13.49 | 13.73 | 32.24 | 65.07 | 36.56 | 56.98 | 57.87 | 54.12 | 31.01 | 7.08 | 23.54 | **20.54** |
| FedNTD G.Lee et al. (2022) | 58.66 | 69.47 | 44.83 | 84.00 | 64.23 | 24.15 | 58.56 | 18.44 | 13.68 | 28.70 | 64.50 | 47.47 | 58.52 | 53.43 | **55.98** | 17.75 | 7.12 | 27.97 | 17.61 |
| Design for Federated Domain Adaptation setting | | | | | | | | | | | | | | | | | | | |
| COPA G.Wu & S.Gong (2021) | 55.17 | 67.05 | 56.55 | 78.33 | 64.27 | 58.93 | 92.20 | 10.49 | 14.90 | 44.13 | 71.61 | 53.74 | 63.12 | 56.60 | 61.26 | 43.06 | 6.69 | 31.26 | 27.00 |
| KD3A H.Feng et al. (2021) | 54.73 | 70.00 | 68.61 | 75.33 | **67.16** | 83.91 | 97.46 | 14.33 | 34.03 | **57.43** | 76.99 | 56.91 | 67.63 | 55.70 | **64.30** | 44.28 | 8.04 | 37.08 | **29.80** |
| Design for Federated Domain Generalization setting | | | | | | | | | | | | | | | | | | | |
| COPA G.Wu & S.Gong (2021) | 57.32 | 66.31 | 48.27 | 70.00 | **60.47** | 33.76 | 47.32 | 13.26 | 15.16 | 27.37 | 59.54 | 35.33 | 56.67 | 57.93 | **52.36** | 21.22 | 5.48 | 19.49 | **15.39** |
| FedGA R.Zhang et al. (2023) | 44.28 | 54.10 | 51.72 | 71.33 | 55.35 | 58.74 | 86.92 | 9.16 | 14.81 | **42.40** | 59.00 | 35.01 | 43.20 | 53.60 | 47.70 | 22.24 | 5.15 | 10.63 | 12.67 |

We measure the average accuracy over all test domains before and after removal,

$$\Delta_i = \bar{A} - \frac{1}{|\mathcal{U}|} \sum_{u \in \mathcal{U}} A_u^{-i},$$

where $\bar{A}$ is the mean accuracy and $A_u^{-i}$ denotes performance on domain $u$ without client $i$. Normalizing the vector $\Delta = (\Delta_1, \ldots, \Delta_M)$ and the weight vector $\alpha$, we define the contribution score

$$\mathcal{C} = \frac{\Delta \cdot \alpha}{\|\Delta\|_2 \, \|\alpha\|_2},$$

so that higher $\mathcal{C}$ indicates closer alignment between actual impact and nominal weights.

**Accuracy Consistency ($\mathcal{V}$).** To evaluate how evenly the model serves all clients, we compute the standard deviation of per-domain accuracies:

$$\mathcal{V} = \sqrt{\frac{1}{|\mathcal{U}|} \sum_{u \in \mathcal{U}} \left(A_u - \bar{A}\right)^2} \times 100\% \, .$$

A smaller $\mathcal{V}$ reflects more uniform performance across heterogeneous client distributions.

# F BENCHMARK SETUP

## F.1 DATA AUGMENTATION

To improve model robustness under data heterogeneity, we apply standard image transformations on each client's local data, implemented via PyTorch routines:

- `RandomCrop(size)`: Crop a random patch of the specified size (e.g., $32 \times 32$ or $224 \times 224$).
- `RandomHorizontalFlip(p)`: Flip images horizontally with probability $p$ (default $p = 0.5$).
- `Normalize(mean, std)`: Scale pixel values to zero mean and unit variance using dataset-specific `mean` and `std` vectors.

## F.2 IMPLEMENTATION DETAILS

**Optimization and Training Protocol.** All methods are evaluated under a common protocol: each client performs $U = 10$ local SGD epochs per communication round, using a batch size of 64, momentum 0.9, and weight decay $10^{-5}$. The learning rate $\eta$ and number of global rounds $E$ vary by task and are specified in Table 11. We choose $E$ such that further rounds yield negligible improvement

Table 11: **Experiments Configuration of different federated scenarios**. Image Size is operated after the resize operation. $|C|$ denotes the classification scale. $|K|$ denotes the clients number. $E$ is the communication epochs for federation. $B$ means the training batch size

| Scenario | Size | $|C|$ | Network $w$ | Rate $\eta$ | $|K|$ | $E$ | $B$ |
|---|---|---|---|---|---|---|---|
| *Label Skew Setting § 4* | | | | | | | |
| Cifar-10 | 32 | 10 | SimpleCNN | 1e-2 | 10 | 100 | 64 |
| Fashion-MNIST | 32 | 10 | SimpleCNN | 1e-2 | 10 | 100 | 64 |
| MNIST | 32 | 10 | SimpleCNN | 1e-2 | 10 | 100 | 64 |
| Cifar-100 | 32 | 100 | ResNet-50 | 1e-1 | 10 | 100 | 64 |
| Tiny-ImageNet | 32 | 200 | ResNet-50 | 1e-2 | 10 | 100 | 64 |
| *Domain Skew / Out-Client Shift Settings § 4* | | | | | | | |
| Digits | 32 | 10 | ResNet-18 | 1e-2 | 4/3 | 50 | 16 |
| PACS | 224 | 7 | ResNet-34 | 1e-3 | 4/3 | 50 | 16 |
| Office Caltech | 224 | 10 | ResNet-34 | 1e-3 | 4/3 | 50 | 16 |
| Office-Home | 224 | 65 | ResNet-34 | 1e-3 | 4/3 | 50 | 16 |

across all algorithms. Experiments are implemented in PyTorch, are seeded for reproducibility and run on NVIDIA RTX 3090 GPUs.

**Model Architectures.** For lightweight benchmarks, we adopt a simple CNN with two $5 \times 5$ convolutional layers (each followed by $2 \times 2$ max-pooling), hereafter called SimpleCNN. Larger datasets use ResNet variants (He et al., 2016). Exact layer counts and input resolutions per scenario are detailed in Table 11.

**Adversary Configurations.** When simulating malicious clients, we vary the fraction of adversaries $\Upsilon \in \{0.2, 0.4\}$. For data-poisoning attacks (SymFlip, PairFlip), the corruption probability is set to $\epsilon = 0.5$. Model-poisoning strategies follow the parameter perturbation schemes described in Section A.2.2.

# G FUTURE WORK

Building on the state of the art, we identify several key challenges for next-generation federated systems:

- **Balancing Generalization and Robustness.** Heterogeneous client data drives the need for broad generalization, yet robustness mechanisms must detect and exclude malicious contributions. When benign clients happen to hold atypical data, they risk being misclassified as attackers, degrading overall performance. Future work should develop joint objectives that preserve legitimate diversity while filtering adversarial behavior.

- **Reconciling Generalization with Fairness.** Optimizing for average accuracy across all clients can obscure poor performance on minority distributions, whereas fairness aims for uniform accuracy regardless of data volume or difficulty. Multi-objective formulations that simultaneously maximize mean accuracy and minimize inter-client variance are needed to avoid this "majority wins" trade-off.

- **Synergies Between Robustness and Fairness.** Accurate contribution metrics underpin both robust outlier rejection and fair reward allocation. By integrating anomaly detection into incentive mechanisms, systems can ensure that low-contribution or malicious clients are neither over-rewarded nor under-penalized, fostering both security and long-term participation.

- **Vertical FL with Generalization, Robustness, and Fairness.** In vertical settings, clients hold complementary feature views of the same entities. Aligning heterogeneous feature sets without leaking private attributes remains an open problem. Moreover, attackers may exploit feature inference or label inference attacks, demanding novel defenses such as secure multi-party computation or homomorphic encryption. Finally, feature-level fairness—ensuring no single view dominates the global model—requires new measures of contribution and bias mitigation.

- **Federating Large Pretrained Models.** Fine-tuning massive foundation models on decentralized data promises strong personalization, but communication costs and intellectual

property concerns pose significant barriers. Research should explore parameter-efficient updates (e.g., adapters, low-rank updates), encrypted or compressed aggregation protocols, and incentive schemes that protect model ownership while enabling collaborative improvement.

- **Enabling Reasoning-Centric Personalization.**

  Current federated learning systems largely optimize for classification or regression tasks, while neglecting reasoning capabilities such as multi-hop inference, commonsense logic, or context-aware question answering. These tasks require richer representations and deeper model understanding—often beyond local training signals. Future research should explore reasoning-aware objectives, knowledge distillation across clients, and hierarchical model structures that enable reasoning patterns to emerge across non-iid data distributions. Additionally, curriculum-based or scaffolded training schedules tailored to client capabilities may allow reasoning modules to be co-learned without centralized supervision.

