# OpenReview forum: "FLAT-Bench: A Federated Learning Benchmark for Adaptation and Trust Evaluation"
_ICLR.cc/2026/Conference — ICLR 2026 Conference Withdrawn Submission_

### Official Review · Reviewer_jAV1 · 2025-10-22

**Soundness:** 1
**Presentation:** 1
**Contribution:** 1
**Rating:** 0
**Confidence:** 1

**Summary:**

This paper proposes FLAT-Bench, a comprehensive benchmark framework for federated learning.  But you just plagiarized a previous federated survey article [1].

[1] Federated Learning for Generalization, Robustness, Fairness: A Survey and Benchmark

**Strengths:**

This paper has put effort in plagiarism.

**Weaknesses:**

1. The differences from existing works, such as TPAMI'24 [1] need clearer articulation; the conceptual definition and necessity of Adaptation as an independent dimension require stronger justification; while the paper covers comprehensive evaluation dimensions, it primarily represents a systematic organization of existing methods and lacks methodological innovation.

2. What is the novelty of the Adaptation and Trust views rather than existing federated survey research?

3. The figures and tables have high information density with suboptimal visualization; the relationship between Generalization in Adaptation and Generalization in Trust is not clearly articulated.


[1] Federated Learning for Generalization, Robustness, Fairness: A Survey and Benchmark

**Questions:**

I have no further questions.

---

### Official Review · Reviewer_Wzqj · 2025-10-24

**Soundness:** 1
**Presentation:** 2
**Contribution:** 2
**Rating:** 2
**Confidence:** 4

**Summary:**

The paper proposes a benchmark for federated learning (FL), that focuses on i) adaptation (meaning client heterogeneity: label and domain shift, as well as out of distribution, OOD, prediction), and ii) trust (robustness against data & model poisoning, against backdoors, and fairness in the sense of equal contributions and accuracy parity). The benchmark currently includes a large number of existing methods, that are evaluated on various standard datasets (eg, CIFAR10 & 100, Fashion MNIST, MNIST, Digits, Office Caltech, PACS) and models (simple CNN, ResNet), depending on the specific task.

**Strengths:**

* I believe a well-implemented and fairly broad benchmark would be valuable for the field: there are currently a wide variety of existing methods mostly designed to tackle one particular issue in FL each, several benchmarks similarly focused on some single performance issue, and quite limited comparisons available that would consistently consider a larger set of methods under various test settings.
* The paper is mostly well-written and easy to read.

**Weaknesses:**

1) As it currently stands, the benchmark seems clearly unfinished (see Questions for more specific comments on this).

2) There is no code available (although the authors do note that they plan to open source the implementation).

**Questions:**

## Questions and comments, mostly in decreasing order of importance:

1) Please include proper details on hyperparameter tuning for each method (eg, grids/limits for the possible hypers, how the hypers are optimized etc.)

2) Include, eg, standard error of the mean or some other variability metric to the results to help evaluate how reliable the results are.

3) It would seem sensible to include at least some datasets with inherent client splits to have a more realistic idea of what heterogeneous data looks like outside simulations. Have you considered adding some?

4) Please fix the Appendices (eg, Appendix B: Hyperparameters is empty, highlights included in the submitted pdf, cf. lines 1362-65: I guess the actual reasoning comparisons were not ready in time for submitting?).

5) FL is often divided into cross-silo vs cross-device types, depending on how many clients, much data & compute the clients are assumed to have etc. Including this (at least in the sense of focusing clearly and explicitly on one of these if not both and doing data splits etc. accordingly) could help make the benchmark better.

6) It seems frankly odd in Table 3 to have a single limitation per method as these do not seem to be on any common scale (is the use of GAN really the single most important limitation for Adversarial alignment, or leaking style cues for client-wise style mixing?). On a similar vein, in Table 4, eg, MultiKrum has limitation "Poor handling of data heterogeneity", while "Trim median", ie, using trimmed median (or mean as written in Table 4?) per dimension has the limitation "Depends on strong mathematical assumptions". Why are these specific limitations the ones chosen here, are the assumptions in MultiKrum somehow clearly weaker than what you need for trimmed median, and does this imply that using trimmed median per feature is clearly better with heterogeneous data than MultiKrum? Overall, it might be better to characterize the methods based on the actual experiments in this paper, not try to put everything on somehow equal seeming footing; there are no doubt clearly better and clearly worse methods for given setups, and it would be very nice to be able see when a given method works and when it does not in such a table.

7) Lines 81-94: this sounds like you would evaluate privacy in the benchmark, which I guess is not true? Also note that this paragraph seems to directly contradict eg lines 43-44 & 72 stating that FL preserves privacy.

8) As a minor presentation comment, it feels a bit weird to have quite lengthy descriptions with image examples of standard datasets in the main text, while far more important seeming things like the general problem formulation are in the Appendix.

---

### Official Review · Reviewer_KDh8 · 2025-10-25

**Soundness:** 1
**Presentation:** 1
**Contribution:** 1
**Rating:** 0
**Confidence:** 5

**Summary:**

This paper proposes a unified federated learning framework, denoted as FLAT-Bench. However, this paper is similar to the survey published on TPAMI 24 [1].

[1] Federated Learning for Generalization, Robustness, Fairness: A Survey and Benchmark.

**Strengths:**

I do not find the strength of this work.

**Weaknesses:**

1. This paper is similar to the paper I mentioned in the summary. What is the main difference to distinguish them?
2. The relationship between  Generalization in Adaptation & Trust is unclear.
3. The figures are almost identical to those in [1].

**Questions:**

I do not have qustions for this type of paper.

---

### Official Review · Reviewer_xvHB · 2025-10-28

**Soundness:** 2
**Presentation:** 2
**Contribution:** 3
**Rating:** 6
**Confidence:** 3

**Summary:**

The paper presents a comprehensive benchmark that consolidates adaptation and trustworthiness challenges in federated learning, offering a structured evaluation framework and empirical comparisons across diverse settings. The taxonomy and experimental design are well-organized, covering key aspects such as cross-client calibration and Byzantine tolerance. Additionally, the differentiation from prior benchmarks could be more explicitly articulated.

**Strengths:**

FLAT-Bench is the first benchmark to unify adaptation and trustworthiness evaluation in FL, providing a holistic taxonomy and standardized protocols.

Extensive experiments cover multiple datasets (e.g., CIFAR-10, PACS), distribution shifts (label/domain skew), and adversarial scenarios (Byzantine/backdoor attacks), ensuring broad applicability.
The paper is well-structured, with clear sections on problem formulation, methodology, and results, supported by concise tables and figures.
The benchmark identifies critical research gaps (e.g., reasoning evaluation, efficiency trade-offs) and outlines actionable future directions.
The classification of methods (e.g., client regularization, server-side adaptation) is logically grounded and aligns with FL challenges.

**Weaknesses:**

While Table 1 summarizes existing benchmarks, the manuscript lacks a detailed comparison of how FLAT-Bench advances beyond them in terms of task formulation or evaluation metrics.

Experiments rely on standardized datasets (e.g., Digits, Office Caltech), with no evidence of validation on real-world, large-scale federated systems.

The impact of individual benchmark components (e.g., metric choices, aggregation strategies) is not analyzed, weakening insights into their necessity.
The paper does not discuss scenarios where evaluated methods underperform severely, such as under extreme heterogeneity or sophisticated attacks.
Contribution and performance fairness metrics (e.g., \(C\) and \(V\)) are introduced but not critically evaluated for robustness across diverse client distributions.
While Byzantine and backdoor defenses are tested, privacy-related threats (e.g., inference attacks) are not incorporated into the trust benchmark.

**Questions:**

Please respond to the Weaknesses.

---

### Note · Authors · 2025-11-12

**Comment:**

We, the authors, wish to withdraw our submission “FLAT-Bench: A Federated Learning Benchmark for Adaptation and Trust Evaluation” as the current version does not meet our quality expectations and requires major revisions.

**Withdrawal Confirmation:**

I have read and agree with the venue's withdrawal policy on behalf of myself and my co-authors.